# Landscape expansion microscopy reveals interactions between membrane and phase-separated organelles

Yinyin Zhuang[1] , Zhao Zhang[1] , Zhipeng Dai[1] , and Xiaoyu Shi[1,2,3]

**Landscape expansion microscopy (land-ExM) is a light microscopy technique that visualizes both the lipid and protein ultrastructural context of cells. Achieving this level of detail requires both superresolution and a high signal-to-noise ratio. Although expansion microscopy (ExM) provides superresolution, obtaining high signal-to-noise images of both proteins and lipids remains challenging. land-ExM overcomes this limitation by using self-retention trifunctional anchors to significantly enhance protein and lipid signals in expanded samples. This improvement enables the accurate visualization of diverse membrane organelles and phase separations, as well as the 3D visualization of their contact sites. As a demonstration, we revealed triple-organellar contact sites among the stress granule, the nuclear tunnel, and the nucleolus. Overall, land-ExM offers a high-contrast superresolution platform that advances our understanding of how cells spatially coordinate interactions between membrane organelles and phase separations.**

## Introduction

Resolving the ultrastructural protein and lipid context of the cell is crucial for understanding how proteins and lipids are spatially assembled for cellular functions. It requires imaging techniques that provide both superresolution below 100 nm and good contrast of proteins and lipids. EM techniques, such as focused ion beam scanning EM (FIB-SEM) and cryo-electron tomography (cryoET), have provided nanometer resolution. They have therefore been the main approaches for imaging of cell context (Grünewald et al. 2003; Narayan and Subramaniam, 2015; Rigort et al., 2012; Xu et al., 2017) and membrane contact sites (Obara et al., 2024; Wu et al., 2018). Correlative light and EM (CLEM) has further combined the strengths of EM in resolution and light microscopy in protein specificity to offer spatially resolved, multimodal insights into cellular ultrastructure (de Boer et al., 2015; Godman et al., 1960; Hauser et al., 2017). Over the past 2 decades, superresolution light microscopy has rapidly narrowed the resolution gap between electron and light microscopy (Balzarotti et al., 2017; Betzig et al., 2006; Gustafsson, 2000; Hell and Wichmann, 1994; Huang et al., 2008; Pavani et al., 2009; Rust et al., 2006). More recently, expansion microscopy (ExM) has emerged as a light microscopy solution to visualize the ultrastructural protein and lipid context (Damstra et al., 2022; Karagiannis et al., 2019, Preprint; Klimas et al., 2023; Mao et al., 2020; M'Saad and Bewersdorf, 2020; Shin et al., 2025; Sun et al., 2021; White et al., 2022).

By physically expanding cells or tissues, ExM allows light microscopes to achieve an effective resolution that is 3- to 20-fold higher than before expansion (Chang et al., 2017; Chen et al., 2015; Damstra et al., 2022; Ku et al., 2016; Li et al., 2022; Shaib et al., 2025; Truckenbrodt et al., 2018; Wang et al., 2024). ExM methods were initially developed for superresolution imaging of targeted proteins (Chen et al., 2015; Chozinski et al., 2016; Ku et al., 2016; Tillberg et al., 2016) and mRNAs (Chen et al., 2016). Over time, recent advancements have introduced innovative approaches for imaging ultrastructural contexts composed of proteins, lipids, DNA, and carbohydrates (Damstra et al., 2022; Karagiannis et al., 2019, Preprint; Klimas et al., 2023; M'Saad and Bewersdorf, 2020; Mao et al., 2020; Shin et al., 2025; Sun et al., 2021; White et al., 2022). These methods created EM-like images on light microscopes, which have slightly lower resolution but are more accessible than electron microscopes. For example, lipid ExM has enabled detailed imaging of lipids by integrating lipid-binding dyes into the ExM protocol (Karagiannis et al., 2019, Preprint; Shin et al., 2025; White et al., 2022). Click-ExM integrates click chemistry with ExM, allowing for labeling and imaging of a wide range of biomolecules, including lipids, glycans, proteins, DNA, and RNA (Sun et al., 2021). Fluorescent labeling of abundant reactive entities resolved protein, carbohydrate, and DNA contexts by covalently staining cells with

---

[1]Department of Developmental and Cell Biology, University of California, Irvine, Irvine, CA, USA;   [2]Department of Chemistry, University of California, Irvine, Irvine, CA, USA;   [3]Department of Biomedical Engineering, University of California, Irvine, Irvine, CA, USA.

Correspondence to Xiaoyu Shi: xiaoyu.shi@uci.edu.

small fluorescent dyes followed by ExM (Mao et al., 2020). Pan-ExM (M'Saad and Bewersdorf, 2020), Ten-fold robust expansion microscopy (TREx) (Damstra et al., 2022), and magnify (Klimas et al., 2023) achieved higher expansion factors for more details of proteins, lipids, or DNA ultrastructures. Chromatin ExM specializes in resolving nanoscale chromatin architecture using metabolic labeling of DNA (Pownall et al., 2023). These methods have been combined with immunostaining to localize targeted proteins on contextual channels, providing an all-optical alternative to CLEM with slightly lower resolution but higher accessibility and matching resolution between specific targets and contextual structures.

Despite the progress achieved with ExM, achieving high signal-to-noise ratios simultaneously in both protein and lipid imaging remains challenging. To optimize the signal-to-noise ratios for both protein and lipid imaging, we have developed a new labeling and anchoring strategy, called landscape ExM (land-ExM). This method builds on our concept of a self-retention trifunctional label for ExM (Shi et al., 2021). The self-retention label reacts with biomolecules, anchors itself to the hydrogel, and can be fluorescently labeled after gelation. In our prior work on label retention ExM (LR-ExM), we demonstrated that antibodies conjugated with self-anchoring trifunctional labels achieved several-fold higher brightness compared with ExM using regular fluorescent antibodies (Shi et al., 2021; Zhuang and Shi, 2023). In this work, we extend this approach by employing self-anchoring probes that not only label and anchor all proteins but also cross-link lipid probes to the hydrogel. This strategy provides bright signals for both protein and lipid channels and simplifies the workflow of contextual ExM.

While the protein channel of land-ExM reveals phase separations—the membraneless compartments, the lipid channel specifies membrane organelles based on their morphologies. By leveraging both channels and combining with immunostaining, land-ExM becomes an efficient tool for identifying contact sites between membrane organelles and phase separations. In this study, we demonstrated membrane and phase separation contact sites involving three structures: the stress granule (SG), the nuclear tunnel, and the nucleolus. This finding illustrates that land-ExM is a powerful tool for the scientific community to investigate the intricate interactions between organelles in cells and tissues.

## Results

### Principle and method development

Fig. 1 A illustrates the workflow of land-ExM, which consists of eight sequential steps: (1) cell fixation, (2) staining lipids, (3) optional immunostaining, (4) anchoring proteins and lipids and staining proteins, (5) gelation, (6) heat denaturation, (7) fluorescence labeling, and (8) expansion. The critical step in achieving a high signal-to-noise ratio for protein and lipid imaging is step (4), where the trifunctional anchor N-hydroxysuccinimide (NHS)-biotin-methacrylate (MA) (Fig. 1 B) simultaneously anchors proteins and the lipid probe, mCLING. NHS-biotin-MA contains three functional groups: (1) NHS ester, which reacts with primary amines in proteins, antibodies, and mCLING containing seven primary amines (Fig. 1 C); (2) MA, which covalently

inserts to the acrylamide polymer during gelation; and (3) biotin, which enables protein imaging by coupling with fluorescent streptavidin. Initially designed for LR-ExM (Shi et al., 2021), this trifunctional probe has been adapted for land-ExM to optimize the signal-to-noise ratio of contextual imaging.

To assess the efficacy of land-ExM in protein imaging, we compared the fluorescence signal from NHS-biotin-MA with that from standard protein dyes, including covalent NHS dyes (e.g., Alexa Fluor 488) and non-covalent stains (e.g., SYPRO Orange). The workflows for these control groups were identical to land-ExM except for steps (4) and (7), where NHS-MA or glycidyl MA (GMA) was used as the protein anchor in step (4), and proteins were stained with NHS-Alexa Fluor 488 (NHS-488) or SYPRO Orange in step (7). For land-ExM, streptavidin conjugated with Alexa Fluor 488 was used to label NHS-biotin-MA in step (7). The reaction between NHS-biotin-MA and streptavidin follows a 1:1 stoichiometry, with each streptavidin molecule conjugated to an average of 0.9 dye molecules. Consequently, each NHS-biotin-MA is converted to about 1 Alexa Fluor 488 molecule in the land-ExM method, which allows for a fair comparison with the control groups.

Our results show that the land-ExM protein image has significantly higher signal-to-noise ratios than ExM using NHS-488 and SYPRO Orange (Fig. 1, D–F). The signal-to-noise ratio of the land-ExM protein image is 8 and 30 times higher than NHS-488 and SYPRO Orange staining, respectively (Fig. 1 G). It is likely because NHS-488 competes with the anchor molecule NHS-MA for primary amines, while NHS-MA consumes primary amines earlier in the workflow than NHS-488. In contrast, NHS-biotin-MA reacts with all primary amines without competition from other steps. In addition, NHS-biotin-MA anchors itself to the hydrogel, while NHS-488 relies on proteins to anchor to the hydrogel indirectly. The drawback of indirect anchoring is that NHS-488 on proteins not anchored to the hydrogel will be washed away during expansion. The self-anchoring strategy using NHS-biotin-MA avoids probe loss and further enhances the signal.

NHS-biotin-MA also anchors the lipid dye mCLING by reacting with its primary amines. Atto 647N fluorescence from anchored mCLING forms the lipid channel for land-ExM. First, we optimized the mCLING staining protocols for the best labeling efficiency of lipids in cultured cells. We found that the working concentration of mCLING is important, and it needs to be optimized for each lot of the product (Fig. S1). Second, we optimized the concentration and incubation duration of NHS-biotin-MA for high-quality lipid imaging in companion with the protein channel. Fig. 1, H–J depicts a 3D stack of protein and lipid images of an expanded mammalian cell with a high signal-to-noise ratio in both channels. The lipid channel (blue) reveals a nuclear tunnel, where the nuclear membrane invaginates deeply through the whole nucleus, while the protein channel (gray) highlights the nucleolus with unique phase separation. The measured lateral resolution of the Airyscan microscope is 138 nm. The 4.0 linear expansion factor of the cells used for Fig. 1 results in an effective lateral resolution of 35 nm. With this resolution and land-ExM's high signal-to-noise ratio, contact sites between the nuclear tunnel and the nucleolus were clearly

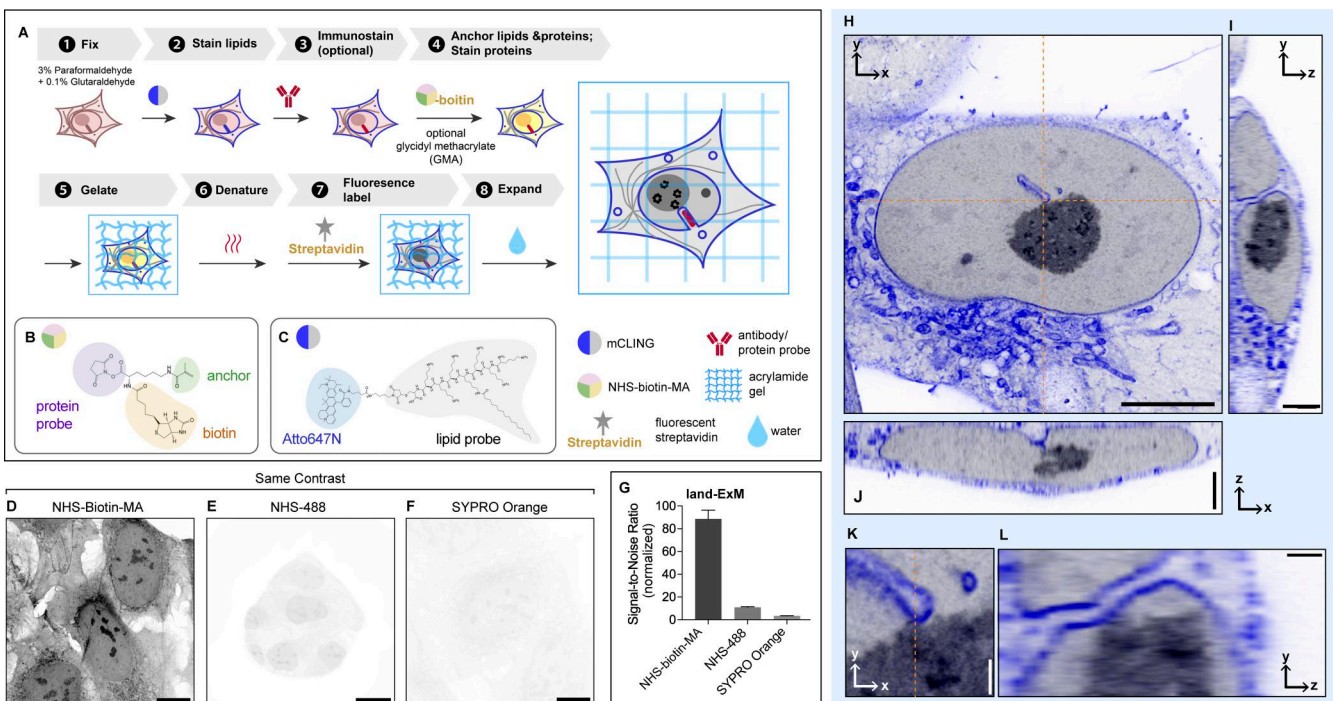

Figure 1. **land-ExM visualizes the protein and lipid context of cells. (A)** Workflow of land-ExM. **(B)** Schematic of NHS-biotin-MA linker. **(C)** Schematic of mCLING. **(D)** land-ExM image of U2OS cells incubated with NHS-biotin-MA linker. Scale bar: 10 μm in pre-expansion unit. Linear expansion factor: 4. **(E)** ExM image of U2OS cells incubated with NHS-MA linker and stained with Alexa Fluor 488 NHS ester dye. Scale bar: 10 μm in pre-expansion unit. Linear expansion factor: 4.2. **(F)** ExM image of U2OS cells incubated with GMA linker and stained with SYPRO Orange. Scale bar: 10 μm in pre-expansion unit. Linear expansion factor: 4.2. **(G)** Bar chart comparing signal-to-noise ratios of protein context images obtained with different ExM methods shown in D–F. The signal-to-noise ratio is calculated as the average pixel value of the area with cells divided by the average pixel value of the area without cells in each image. Each bar represents the mean ± standard error of more than 10 cells. **(H–J)** Different views of land-ExM images of a breast cancer cell, UCI082014, stained with mCLING for lipid content. The orange dashed lines in H show where the orthogonal views (I and J) align. Scale bar: 5 μm (H), 2 μm (I and J) in pre-expansion unit. Linear expansion factor: 3.8. **(K)** Magnified images of H. **(L)** Magnified images of I. The orange dashed line in K shows where the orthogonal view (L) aligns. Scale bar: 0.5 μm in pre-expansion unit. Linear expansion factor: 3.8. All images were taken with an Airyscan microscope. Images D–F were adjusted to the same contrast. Image in D is also shown in Fig. S3 C.

visualized (Fig. 1, K and L). This observation is consistent with previous findings using EM (Bouteille et al., 1979; Malhas et al., 2011). Compared with EM, land-ExM's faster speed, 3D imaging, and multiplexity will enable a more statistical understanding of the interactions between the nuclear tunnel and the nucleolus. We explored the frequency and functions of the nuclear tunnel–nucleolus interaction with land-ExM in a recent preprint (Zhuang et al., 2024, *Preprint*).

As NHS-biotin-MA reacts to both proteins and mCLING, there is a potential risk of cross-contamination between protein and lipid signals. To avoid cross talk, we attempted an alternative workflow that anchors and stains proteins with NHS-biotin-MA before the addition of mCLING (Fig. S2 A). GMA is subsequently used as an additional anchor to retain the mCLING signals in gel (Fig. S2 A, step 5). As expected, the NHS channel showed distinct fluorescence patterns from the mCLING channel (Fig. S2 B). We compared the result of this cross talk–free workflow (Fig. S2 B) with the original workflow (Fig. S2 C), finding that the risk of cross-contamination in the original workflow is negligible. It is due to the significantly lower abundance of mCLING than native proteins. Therefore, for typical cell lines and tissues, the original workflow (Fig. 1 A) should be reliable. But for samples exceptionally rich in lipid, we recommend the cross talk–free land-ExM

workflow (Fig. S2 A). For users focusing on lipid context but with a low requirement for protein imaging, commercial NHS-MA or GMA (Cui et al., 2023) can serve as an alternative to NHS-biotin-MA (Fig. 1 A, step 4), and NHS dyes can be used for protein fluorescent labeling in step 7. This will yield an equally high signal-to-noise ratio in lipid imaging following our instructions in the Materials and methods section, but with significantly dimmer signals in the protein channel (Fig. 1 G).

### land-ExM is compatible with proteinase K digestion

Another advantage of land-ExM is its compatibility with proteinase K digestion. Proteinase K digestion was widely used for antibody-based ExM techniques (Chen et al., 2015; Chozinski et al., 2016; Tillberg et al., 2016), which usually results in a higher expansion factor and less distortion than heat denaturation. However, most current ExM methods for protein and lipid ultrastructural imaging use heat denaturation (Damstra et al., 2022; Karagiannis et al., 2019, *Preprint*; M'Saad and Bewersdorf, 2020; Shin et al., 2025), and are not well compatible with proteinase K digestion. It is because these methods rely on native proteins or antibodies to retain the NHS ester dye in the hydrogel. If proteinase K is used, it can fragment proteins and antibodies and cause loss of NHS ester dye on the fragments that

do not cross-link to the hydrogel. However, land-ExM avoids this problem because all protein and lipid probes are covalently anchored through the NHS-biotin-MA. This advantage made land-ExM compatible with proteinase K digestion. Fig. S3 A showed the workflow of the proteinase K version of land-ExM. As expected, land-ExM using proteinase K retained protein and lipid signals after expansion (Fig. S3, B and D). Subcellular structures identified in the heat denaturation land-ExM workflow are identifiable using the proteinase K land-ExM workflow (Fig. S3 C). The lipid signal after proteinase K digestion (Fig. S3 D) is dimmer than that obtained using heat denaturation (Fig. S3 E), but it can still reveal the signature morphology of most membrane organelles.

### land-ExM enhances protein and lipid signals of TREx and pan-ExM

We applied the labeling and anchoring strategy of the land-ExM to TREx (Damstra et al., 2022) and pan-ExM (M'Saad and Bewersdorf, 2020), which are existing ExM techniques for ultrastructural protein and lipid context imaging using a higher expansion factor (7–20 times). By simply switching the gelation monomer to that of TREx at step 5 in Fig. 1 A, we achieved an expansion factor of 7.0. We term this land-ExM variant land-TREx (Fig. 2, B and E). Compared with TREx (Fig. 2, C and F), land-TREx showed a fivefold increase in signal-to-noise ratio in the protein channel and a twofold increase in the lipid channel (Fig. 2, D and G). This improvement accounts for the use of NHS-biotin-MA. To integrate land-ExM with pan-ExM (land-pan-ExM), we introduced NHS-biotin-MA right after cell fixation (Fig. 2 A). In the original pan-ExM workflow, the acrylamide and formaldehyde incubation step was designed to anchor proteins to the hydrogel and to prevent inter-protein cross-linking. This step consumes primary amines of proteins, which compete with the reaction between NHS ester dye and primary amine at a later step and thus hinder the protein labeling efficiency. However, land-pan-ExM avoids this competition by labeling and anchoring the proteins through NHS-biotin-MA beforehand. As a result, the signal-to-noise ratio of the protein channel of land-pan-ExM was two times higher than that of pan-ExM (Fig. 2, H–J). In addition, land-pan-ExM showed a threefold increase in lipid signal compared with pan-ExM. (Fig. 2, K–M). This is because NHS-biotin-MA anchors mCLING to the hydrogel in addition to formaldehyde. Overall, we showed that land-ExM is compatible with other ExM techniques and enhances protein and lipid signals.

### land-ExM visualizes membrane organelles and phase separation

land-ExM can visualize membrane structures and phase-separated structures based on their morphologies and locations in both protein and lipid channels. Here, we define phase separation as membraneless protein condensates. Fig. 3, A–G display a gallery of land-ExM protein images of phase-separated compartments, such as nucleoli (Fig. 3 A), nuclear bodies (Fig. 3 B), and SGs (Fig. 3 C). Within the nucleus, chromatin (Fig. 3 D) and nuclear pore complexes (NPCs) (Fig. 3 E) were resolved, while in the cytoplasm, mitochondria (Fig. 3 F), and cytoskeleton

(Fig. 3 G) were visualized. A contributor to the cytoskeleton structure in the protein channel is actin, which was confirmed with phalloidin staining (Fig. S6). Similarly, lipid land-ExM imaging distinguishes membrane structures (Fig. 3, H–P), such as lipid vesicles (Fig. 3 I), mitochondria (Fig. 3 J), filopodia (Fig. 3 K), nuclear membrane invaginations (Fig. 3 L), and Golgi apparatuses (Fig. 3 M). Even the trans and cis faces of a Golgi apparatus can be discernible based on lipid curvature (Fig. 3 M).

In addition to the protein and lipid channels, immunostaining can be added to specify the organelles further (Fig. S4, Fig. S5, and Fig. S6). For example, using anti-Lamp2 and anti-clathrin antibodies, lysosomes and clathrin-coated pits can stand out from other lipid vesicles (Fig. S4). The 3D land-ExM lipid image can reveal the whole cell membrane with intricate structures (Fig. 3 N), such as filopodia, cytonemes (Fig. 3 O), and potential exosomes (Fig. 3 P). With bright signals in protein, lipid, and immunostaining, land-ExM is optimized for discovering interactions between specific membrane organelles and phase separation.

### Triple-organellar contact sites among the SG, the nuclear tunnel, and the nucleolus

Studying organellar contact sites is important because these regions are not just passive points of membrane proximity but are active hubs for communication, coordination, and metabolic regulation inside cells. The region between two organelles that come into proximity, typically separated by only 10–80 nm, is considered a contact site (Raimondi et al., 2023; Scorrano et al., 2019). Multicolor 3D land-ExM with lipid, protein, and antibody channels offers the specificity and resolution needed to identify organellar contact sites between organelles, including both membrane and membraneless organelles. We measured the distance between the membranes of two organelles or between a membrane and the edge of a phase-separated organelle as a metric to discriminate a contact site.

As a demonstration, we investigated SGs, which suppress mRNA translation by sequestering mRNAs into phase-separated compartments in the cytoplasm. To form SGs, we treated the cells in 500 μM sodium arsenite for 20 and 60 min, respectively, and compared them with the ones without treatment (Fig. S7). We immunostained the cells with SG marker G3BP1 and visualized it in the context of proteins and lipids using the multicolor 3D land-ExM. When the cells were not treated with sodium arsenite, G3BP1 diffused across the cell (Fig. S7, A and D). When the cells were incubated in a sodium arsenite solution, G3BP1 condensates appeared (Fig. S7, B, C, E, and F). And when we removed the sodium arsenite, the G3BP1 condensates disappeared. These results validate that the G3BP1 condensates are caused by stress, and G3BP1 can be used as a SG marker, which is consistent with how sodium arsenite was used in previous research to introduce SGs (Wheeler et al., 2016). Surprisingly, some SGs appeared near the nucleolus (boxed in Fig. 4 A and magnified in Fig. 4, B–E) instead of the cytoplasm (Fig. S7 G). We asked whether these two phase-separated organelles directly interact in the nucleus. The answer is no when considering the lipid context. The lipid channel showed that these small SGs were localized inside nuclear tunnels, separated from the nucleolus by

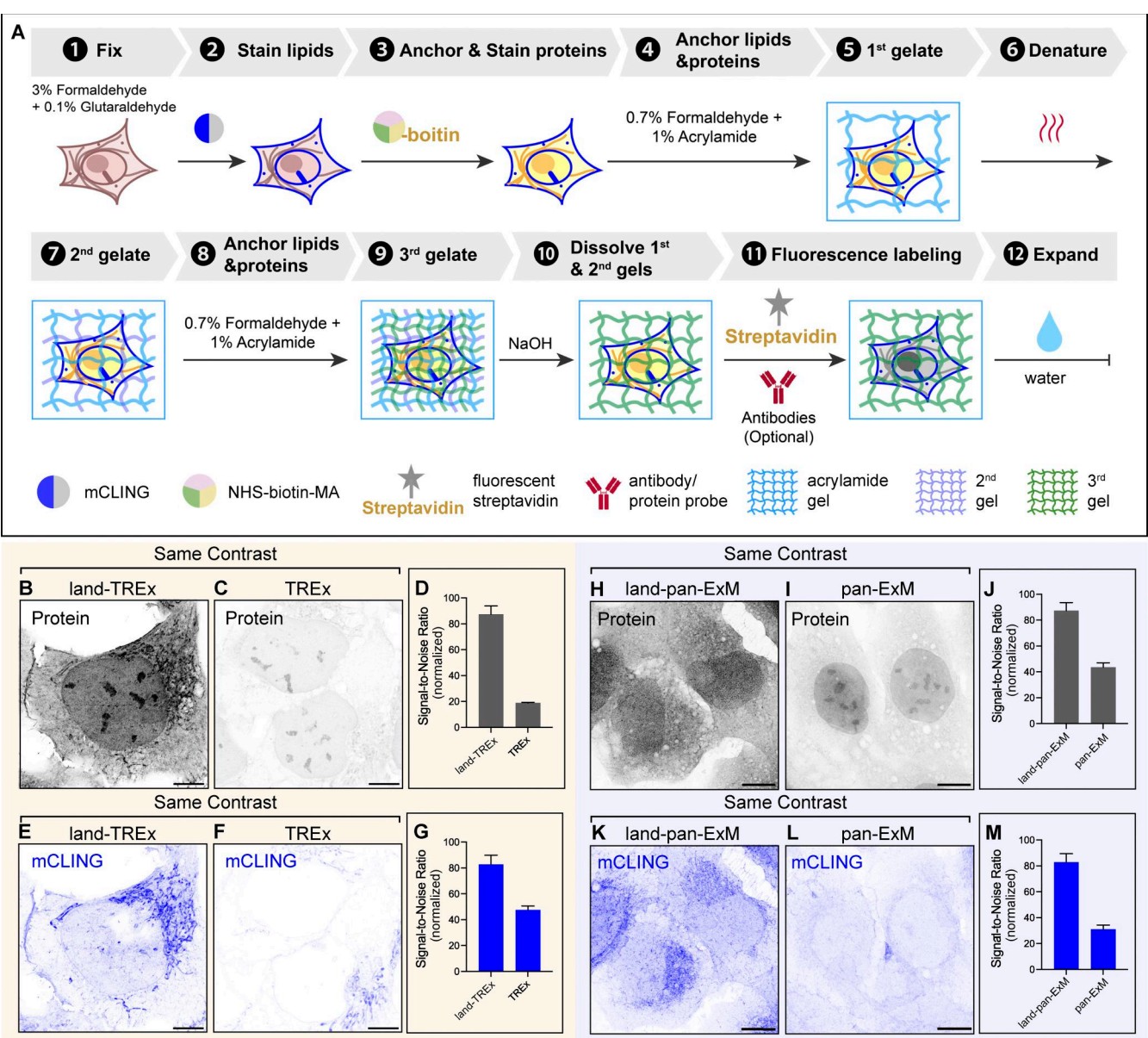

Figure 2. **land-ExM labeling and anchoring strategies improve the signal of TREx and pan-ExM. (A)** Workflow of land-pan-ExM, which only replaces the labeling strategy of pan-ExM with the labeling strategy of land-ExM. **(B)** land-TREx protein channel of U2OS cells, where proteins were labeled and anchored with NHS-biotin-MA. Scale bar: 5 µm in pre-expansion unit. Linear expansion factor: 7. **(C)** TREx protein channel of U2OS cells, where proteins were anchored with acryloyl-X SE and stained with Alexa Fluor 488 NHS ester. Scale bar: 5 µm in pre-expansion unit. Linear expansion factor: 7. **(D)** Bar chart comparing the signal-to-noise ratio of the protein channel in land-TREx and TREx. The signal-to-noise ratio is calculated as the average pixel value of the area with cells divided by the average pixel value of the area without cells in each image. Each bar represents the mean ± standard error of more than 20 cells. **(E)** land-TREx lipid channel of U2OS cells, where lipids were labeled by mCLING and anchored with NHS-biotin-MA. Scale bar: 5 µm in pre-expansion unit. Linear expansion factor: 7.0. **(F)** TREx lipid channel of U2OS cells, where lipids were anchored with acryloyl-X SE and stained with mCLING. Scale bar: 5 µm in pre-expansion unit. Linear expansion factor: 7.0. **(G)** Bar chart comparing the signal-to-noise ratio of the lipid channel of land-TREx and TREx. The signal-to-noise ratio is calculated as the average pixel value of the area with cells divided by the average pixel value of the area without cells in each image. Each bar represents the mean ± standard error of more than 20 cells. **(H)** land-pan-ExM protein channel of U2OS cells, where proteins were labeled and anchored with NHS-biotin-MA. Scale bar: 5 µm in pre-expansion unit. Linear expansion factor: 12.0. **(I)** Pan-ExM protein channel of U2OS cells labeled with Alexa Fluor 488 NHS ester. Scale bar: 5 µm in pre-expansion unit. Linear expansion factor: 12.0. **(J)** Bar chart comparing the signal-to-noise ratio of the protein channel in land-pan-ExM and pan-ExM. The signal-to-noise ratio is calculated as the average pixel value of the area with cells divided by the average pixel value of the area without cells in each image. Each bar represents the mean ± standard error of more than 20 cells. **(K)** land-pan-ExM lipid channel of U2OS cells, where lipids were stained following the workflow (A). Scale bar: 5 µm in pre-expansion unit. Linear expansion factor: 12.0. **(L)** Pan-ExM lipid channel of U2OS cells labeled with mCLING. Scale bar: 5 µm in pre-expansion unit. Linear expansion factor: 12.0. **(M)** Bar chart comparing the signal-to-noise ratio of the lipid (mCLING) channel in land-pan-ExM and pan-ExM. The signal-to-noise ratio is calculated as the average pixel value of the area with cells divided by the average pixel value of the area without cells in each image. Each bar represents the mean ± standard error of more than 20 cells. All images were taken with an Airyscan microscope.

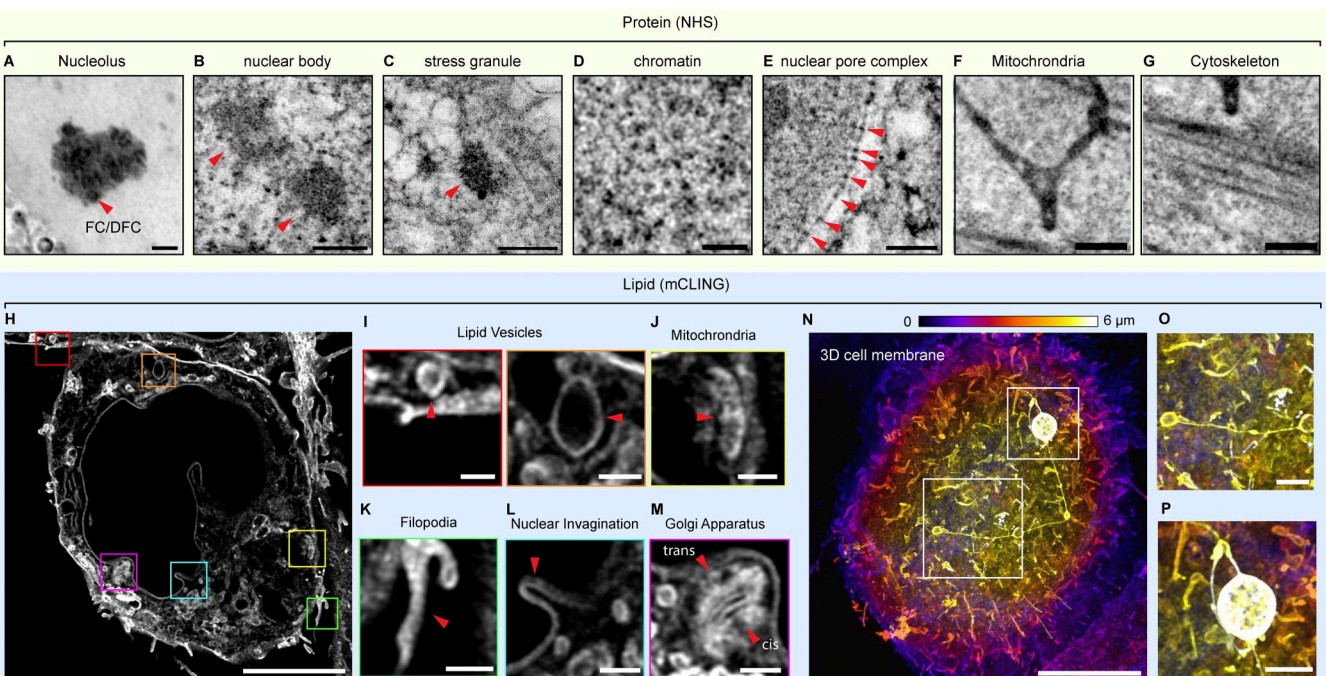

Figure 3.  **land-ExM visualizes phase-separated and membrane organelles. (A–G)** land-ExM protein images of membraneless phase separation structures. The proteins were labeled with NHS-biotin-MS and after gelation stained with streptavidin-Alexa Fluor 488. **(A)** land-ExM protein image of nucleoli in a U2OS cell. Red arrowheads indicate the fibrillar center (FC) or dense fibrillar component (DFC) of the nucleolus. Scale bar: 1 µm in pre-expansion unit. Linear expansion factor: 4.0. **(B)** land-ExM protein image of nuclear bodies of breast cancer cell, UCI082014. Red arrowheads indicate the nuclear bodies. Scale bar: 1 µm in pre-expansion unit. Linear expansion factor: 4.2. **(C)** land-ExM protein image of SGs of a U2OS cell treated with NaAsO$_2$ for 20 min. The red arrowhead indicates a SG. Scale bar: 1 µm in pre-expansion unit. Linear expansion factor: 4.0. **(D)** land-ExM protein image of chromatin of a breast cancer cell. Scale bar: 500 nm in pre-expansion unit. Linear expansion factor: 4.2. **(E)** land-ExM protein image of NPCs of a breast cancer cell. Scale bar: 1 µm in pre-expansion unit. Linear expansion factor: 4.2. **(F and G)** land-ExM protein images of mitochondria and cytoskeleton of a U2OS cell. Scale bar: 1 µm in pre-expansion unit. Linear expansion factor: 4.0. **(H–P)** land-ExM lipid images of membrane structures. The lipids were labeled with mCLING-Atto647N. (H) land-ExM lipid image of breast cancer cell. Scale bar: 5 µm in pre-expansion unit. Linear expansion factor: 4.0. (I–M) magnified images of H showing different membrane structures: lipid vesicles (I), mitochondria (J), filopodia (K), nuclear invagination (L), and Golgi apparatus (M). Scale bar: 1 µm (I–M) in pre-expansion unit. (N) 3D land-ExM lipid image of a breast cancer cell after maximum intensity projection, showing the cell membrane. Color-coded by the z-dimension slices from bottom to top. Color bar: purple to white: 0–6 µm in pre-expansion unit. Scale bar: 5 µm in pre-expansion unit. Linear expansion factor: 4.0. (O and P) magnified images of N showing detailed structures of the cell membrane. Scale bar: 1 µm in pre-expansion unit. All images were taken with an Airyscan microscope.

the nuclear membrane (Fig. 4, F–K). The nuclear tunnel forms two contact sites on its cytoplasmic side and nucleoplasmic side (Fig. 4, G and H). It contacts the SG on the cytoplasmic side and the nucleolus on the nucleoplasmic side. This spatial relationship was commonly seen in nuclear tunnels of cells under stress in our experiments. We examined 114 nuclear tunnels in more than 20 cells, finding 83% of the tunnels contain SGs (Fig. 4 R). Among the tunnels containing SGs, 60% contact nucleoli (Fig. 4 S). The triple-organelle interactions may result in efficient reduction of mRNAs as a response to stress. We will discuss the potential mechanisms in the discussion section. Furthermore, we quantitatively examined whether the sodium arsenite treatment alters the nuclear tunnels. We found that U2OS cells can have 1–10 nuclear tunnels per nucleus, with an average number of five tunnels per nucleus (Fig. S7 H). The diameter of the nuclear tunnels ranges from 140 to 400 nm, with an average of around 250 nm (Fig. S7 I). Sodium arsenite treatment had no significant effect on the number or size of nuclear tunnels (Fig. S7, H and I).

We also observed that the ER, as a part of the nuclear membrane of nuclear tunnels, was adjacent to the SGs. In our four-color land-ExM images, which captured lipids, proteins, and immunostained SG and ER markers (Fig. 4 L), the ER was found adjacent to SGs (Fig. 4, M–P). ER–SG contacts inside nuclear tunnels were frequently observed in cells under stress in our experiments. All four nuclear channels shown in Fig. 4 Q displayed contact sites between SGs and the ER. Previous studies reported the contact between SGs and ER in the cytosol, which contributes to the SG's formation (Liu et al., 2024; Nicchitta, 2024; Pincus and Oakes, 2024) and the ER's response to stress (Liu et al., 2024). Our observations in nuclear tunnels highlight parallels between the SG–ER interaction within nuclear tunnels and in the cytoplasm. Our land-ExM imaging shows that the nuclear tunnels provide confined space that enables contact between the membrane and phase-separated organelles, including the ER, SGs, and nucleoli. Our recent study demonstrated that nuclear tunnels regulate ribosome biogenesis by interacting with the nucleolus (Zhuang et al., 2024, *Preprint*). The triple contact among the SG, the nuclear tunnel, and the nucleolus may indicate more roles of nuclear tunnels under stress.

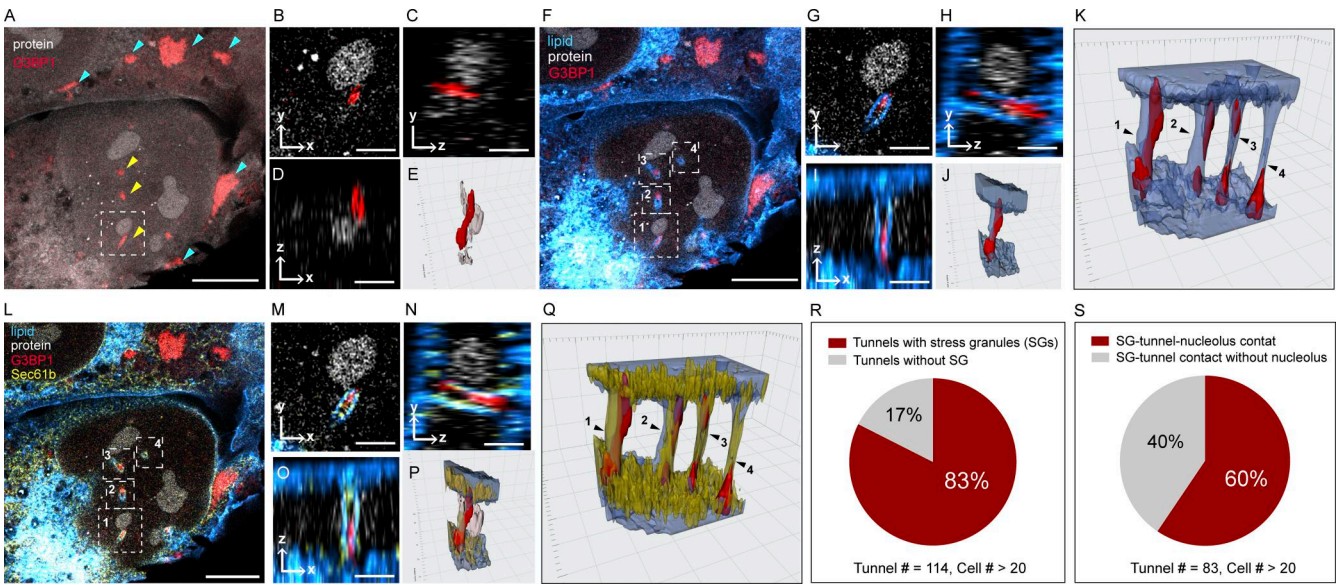

Figure 4. **The nuclear tunnel forms a triple-organellar contact site that includes the SG, the nucleolus, and itself. (A)** land-ExM protein (gray) image of U2OS cells immunostained with anti-G3BP1 (red) antibody. Cells were treated with $NaAsO_2$ for 1 h. Scale bar: 5 µm in pre-expansion unit. Linear expansion factor: 4. **(B–D)** Different views of SG in the white dashed box of A. Scale bar: 1 µm in pre-expansion unit. **(E)** 3D rendering of SG in the white dashed box of A. In the reference grid, the spacing of major and minor tick marks is 0.5 and 0.1 µm in pre-expansion unit. **(F)** land-ExM protein (gray) and lipid (blue) image of U2OS cells immunostained with anti-G3BP1 (red) antibody. Cells were treated with $NaAsO_2$ for 1 h. Scale bar: 5 µm in pre-expansion unit. Linear expansion factor: 4. **(G–I)** Different views of SG in the white dashed box 1 of F. Scale bar: 1 µm in pre-expansion unit. **(J)** 3D rendering of SG in the white dashed box 1 of F. In the reference grid, the spacing of major and minor tick marks is 0.5 and 0.1 µm in the pre-expansion unit. **(K)** 3D rendering of SGs in the white dashed box 1–4 of F. In the reference grid, the spacing of major and minor tick marks is 0.5 and 0.1 µm in the pre-expansion unit. **(L)** land-ExM protein (gray) image of U2OS cells immunostained with anti-G3BP1 (red) and anti-Sec61b (yellow) antibodies. Cells were treated with $NaAsO_2$ for 1 h. Scale bar: 5 µm in pre-expansion unit. Linear expansion factor: 4. **(M–O)** Different views of SG in the white dashed box 1 of L. Scale bar: 1 µm in pre-expansion unit. **(P)** 3D rendering of SG in the white dashed box 1 of L. In the reference grid, the spacing of major and minor tick marks is 0.5 and 0.1 µm in pre-expansion unit. **(Q)** 3D rendering of SGs in the white dashed box 1–4 of L. In the reference grid, the spacing of major and minor tick marks is 0.5 and 0.1 µm in pre-expansion unit. **(R)** Pie chart of nuclear tunnels with or without SGs. Total tunnels analyzed: 114. **(S)** Pie chart of SG-filled nuclear tunnels that contact nucleoli versus those that do not. Total tunnel analyzed: 83. All images were taken with an Airyscan microscope. The cell shown in A, F, and L is also shown in Fig. S7 F.

## Discussion

### Advantages of land-ExM

In this study, we developed a new ExM approach, land-ExM, that significantly enhances the signal-to-noise ratio in imaging protein and lipid ultrastructure. A key innovation in our method is the use of the trifunctional anchor NHS-biotin-MA, which simultaneously anchors proteins and the lipid probe mCLING to the hydrogel. This self-anchoring strategy results in signal-to-noise ratios significantly higher than those achieved with conventional protein dyes such as NHS-488 or SYPRO Orange. The self-anchoring strategy is compatible with both heat denaturation and proteinase K digestion, enabling the integration of land-ExM with most ExM techniques. In this work, we demonstrated on the standard x4 ExM, TREx, and pan-ExM.

Another strength of land-ExM is its compatibility with targeted protein imaging techniques. These include immunostaining, fluorescent proteins, and self-labeling tags. Multicolor land-ExM imaging of targeted molecules, lipids, and proteins enables the precise localization of molecules within the 3D architectures formed by interacting organelles and protein complexes. The combination of contextual and targeted imaging is streamlined in land-ExM, as shown in Fig. 1 A. All imaging channels are captured on the same microscope, ensuring similar resolution and straightforward alignment of all channels.

Compared with CLEM, land-ExM as well as other contextual ExM methods are more accessible, cost-effective, and faster. Its affordability and ease of use make it an attractive option for laboratories with limited resources.

Collectively, these advances in anchoring efficiency, signal brightness, multicolor imaging, and superresolution capability position land-ExM as a powerful method for the detailed investigation of protein and lipid ultrastructure in biological systems.

### Limitations of land-ExM and solutions

Despite its significant advantages, land-ExM has several limitations. While its resolution surpasses that of conventional light microscopy, it is still lower than EM techniques such as cryoET, FIB-SEM, and transmission EM. This limitation hinders the method's ability to resolve fine structural details below 10 nm, such as distinguishing between the inner and outer nuclear membranes. It is technically possible to push the resolution of land-ExM slightly beyond 10 nm by imaging with single-molecule localization microscopes, like STORM (Shi et al., 2021) and MINFLUX (Balzarotti et al., 2017; Schmidt et al., 2021), or employing more swellable hydrogel (Chang et al., 2017; Wang et al., 2024; Shaib et al., 2025). However, the ultimate resolution that ExM can achieve is limited by the pore size of the hydrogel, and

the nanoscale distortion caused by expansion must be carefully examined.

land-ExM is not a label-free technique, and so the contextual information it provides can be influenced by the choice of labeling probes. For protein staining, NHS ester probes give a stronger signal to lysine-rich proteins, potentially introducing bias in the representation of cellular components. To address this, alternative labeling chemistries can be employed. For example, SYPRO Orange, which interacts with the hydrophobic regions of proteins exposed after denaturation, offers complementary insights. Similarly, for lipid staining, different probes exhibit varying labeling efficiency. mCLING is well retained by membranes after fixation and permeabilization, while styryl (FM) dyes are largely lost (Revelo et al., 2014). It can therefore be used in combination with immunostaining, meeting the needs of ExM. Probes specifically developed for ExM, such as those from the Boyden lab, offer improved fluorescence retention for lipid imaging (Shin et al., 2025).

Another challenge is the potential for fixation artifacts, as land-ExM relies on chemical fixation. Fixation conditions optimized for one biomolecule may compromise the labeling efficiency of another. For example, lipid staining often requires glutaraldehyde fixation, which can mask certain protein epitopes and inhibit their immunostaining. These limitations restrict the range of experiments that can be performed using land-ExM. In addition, harsh chemical fixation can cause distortion of cellular ultrastructure at the sub-micrometer scale. One promising alternative, highlighted in recent protocols (Laporte et al., 2022), is cryofixation, which more effectively preserves native cellular structures. To assess the subcellular distortion in expanded samples, quantitative measurement of sub-micrometer distortion should be acquired for ExM experiments, particularly when using a new protocol or expanding a different organelle. GelMAP (Damstra et al., 2023), along with methods that use fiducial markers (Bianchini et al., 2021; Scheckenbach et al., 2020; Vanheusden et al., 2020), offers a solution for measuring local distortions in 2D. However, a technical gap still exists for the expansion factor mapping of whole cells in 3D.

### The nuclear tunnel organizes membrane-phase separation contact

We recently reported that nuclear tunnels can activate the nucleoli they contact, leading to increased ribosomal subunit production in nucleoli (Zhuang et al., 2024, *Preprint*). These subunits are then exported through NPCs located on the nuclear tunnels, thereby elevating mRNA translation in the cell. In this study, we observed SGs adjacent to nucleoli, separated by these nuclear tunnels, in cells under stress. As SGs primarily function to silence mRNA translation by sequestering mRNAs within their phase-separated environments, the close proximity of SGs to nucleoli via nuclear tunnels suggests a functional link. Our hypothesis is that by localizing SGs in the nuclear tunnels, the cell creates a regulatory microenvironment that can capture freshly exported mRNAs and ribosomal subunits—thereby reducing translation. This arrangement may enable cells to efficiently coordinate ribosome production, mRNA sequestration, and protein synthesis in response to stress.

While G3BP1's classic role involves SG assembly, it also contributes to membrane repair (King et al., 2025). For example, G3BP1 is reported to repair lysosomes through a "plugging" mechanism (Bussi et al., 2023). This repair activity positions G3BP1 as a crucial integrator of cellular stress responses across both membraneless and membrane-bound compartments. It is possible that the localization of G3BP1 in the nuclear tunnels is related to these prior studies.

### Conclusion

land-ExM represents a significant advancement in superresolution microscopy by combining contextual ExM imaging with a self-retention trifunctional labeling strategy. This approach enables high signal-to-noise detection of both proteins and lipids while being compatible with immunostaining. In turn, land-ExM provides detailed insights into complex spatial relationships between membrane-bound and phase-separated structures. Notably, it has uncovered contact sites involving three structures: the SG, the nuclear tunnel, and the nucleolus. These results offer new perspectives on organelle–organelle interactions.

With its superresolution, compatibility with existing ExM protocols, and greater accessibility relative to EM, land-ExM is a practical yet powerful solution for investigating the molecular landscape of cells. By enabling researchers to visualize the protein and lipid context in cells and tissues on light microscopes, land-ExM can accelerate discoveries in organellar interactions and structure-function relationships. Future efforts to enhance its resolution, fixation methods, and labeling approaches will likely further expand its impact and applicability.

## Materials and methods

Table 1 shows key resources.

### Cell culture and treatment

All cell lines were maintained in humidified incubator with 5% $CO_2$ at 37°C. U2OS cells (cat#HTB-96; ATCC) were cultured in Mccoy's 5A medium (Cat#16600082; Gibco) supplemented with 10% Fetal Bovine Serum (cat#10082147; Gibco) and 1% penicillin-streptomycin-amphotericin B (cat#A5955; Sigma-Aldrich). UCI082014 breast cancer cells were cultured in DMEM-high glucose supplemented with GlutaMAX, 10% Fetal Bovine Serum, and 1% penicillin-streptomycin-amphotericin B. All cell lines were tested mycoplasma-free using the MycoStrip-Mycoplasma Detection Kit (cat# rep-mysnc-50; InvivoGen) and used at < 10 passages from thaw. UCI082014 cell (Woytash et al., 2025) was a gift from Dr. Olga V Razorenova at the University of California, Irvine, CA, USA.

To induce SGs in the U2OS cells, 500 µM sodium arsenite (cat#AA41533AP; Thermo Fisher Scientific) was added to the culture medium and incubated with cells at 37°C for 20 min to 1 h. Cells were quickly fixed after the sodium arsenite treatment.

### Optimization of lipid staining with mCLING

We found that the mCLING-Atto 647N purchased may have batch-to-batch variations in purity or dye-to-mCLING ratio, which resulted in different lipid labeling efficiencies (Fig. S1).

Table 1.  **Key resources table**

| Materials and reagents | Source | Identifier |
|---|---|---|
| Antibody | | |
| LR-ExM anti-FITC-phalloidin anti-mouse two color kit | Cytoseen | LR-RBF-MD-025 |
| G3BP1 antibody (TT-Y) | Santa Cruz Biotechnology | sc-81940 |
| Sec61b (D5Q1W) Rabbit mAb | Cell Signaling Technology | 14648 |
| Anti-TOMM20 antibody [EPR15581-54] | Abcam | ab186735 |
| LAMP-2 (H4B4) | Santa Cruz Biotechnology | sc-18822 |
| Anti-clathrin heavy chain antibody | Abcam | ab21679 |
| Fluorescein Phalloidin | Invitrogen | F432 |
| Fluorescein/Oregon Green Polyclonal Antibody, Alexa Fluor 488 | Invitrogen | A11096 |
| LR-ExM anti-rabbit antibody-digoxigenin trifunctional anchor and anti-digoxigenin dye kit | Cytoseen | LR-RD488-050 |
| LR-ExM anti-mouse antibody-digoxigenin trifunctional anchor and anti-digoxigenin dye kit | Cytoseen | LR-MD568-050 |
| AffiniPure Goat Anti-Rabbit IgG (H+L) | Jackson ImmunoResearch | 111-005-144 |
| AffiniPure Goat Anti-Mouse IgG (H+L) | Jackson ImmunoResearch | 115-005-166 |
| DyLight 405 Streptavidin | Jackson ImmunoResearch | 016-470-084 |
| Alexa Fluor 488 Streptavidin | Jackson ImmunoResearch | 016-540-084 |
| Protein and lipid labeling reagents | | |
| land-ExM NHS-biotin-MA and Streptavidin-488 kit | Cytoseen | LD-P488-050 |
| land-ExM NHS-biotin-MA and Streptavidin-405 kit | Cytoseen | LD-P405-050 |
| mCLING-Atto 647N | Synaptic Systems | 710 006AT647N |
| Alexa Fluor 488 NHS Ester (Succinimidyl ester) | Invitrogen | A20000 |
| Alexa Fluor 568 NHS Ester (Succinimidyl ester) | Invitrogen | A20003 |
| Other reagents and supplies | | |
| SYLGARD 184 Silicone Elastomer Kit | Ellsworth | 2137054 |
| 12-mm, no. 1 cover glass | Bellco Glass | 1943–10012A |
| Paraformaldehyde | Electron Microscopy Sciences | 50980488 |
| Formaldehyde | Sigma-Aldrich | F8775 |
| Aqueous glutaraldehyde EM Grade 10% | Electron Microscopy Sciences | 16120 |

Table 1.  **Key resources table (*Continued*)**

| Materials and reagents | Source | Identifier |
|---|---|---|
| Triton X-100 | Sigma-Aldrich | 9002-93-1 |
| Bovine serum albumin | Thermo Fisher Scientific | BP9703100 |
| N-Succinimidyl methacrylate (NHS-MA) | Fisher Scientific | S08125G |
| Glycidyl methacrylate (GMA) | Sigma-Aldrich | 151238 |
| Sodium acrylate, 25 g | Santa Cruz Biotechnology | sc-236893 |
| Acrylamide | Sigma-Aldrich | A9099-100 G |
| N,N'-(1,2-Dihydroxyethylene)bisacrylamide | Sigma-Aldrich | 294381-5G |
| N,N'-Methylenebisacrylamide | Sigma-Aldrich | M7279-25 G |
| N,N,N',N'-Tetramethylethylenediamine | Sigma-Aldrich | T9281-100 Ml |
| Ammonium persulfate | Sigma-Aldrich | A3678-25 G |
| Sodium Dodecyl Sulfate | Thermo Fisher Scientific | BP166-500 |
| 1 M Tris HCl Buffer, pH 7.5 | Invitrogen | 15567027 |
| Proteinase K | Sigma-Aldrich | P4850-5Ml |

The powder of mCLING-atto647N should be blue in color. Be aware that if the powder is received in white, such mCLING has limited lipid staining capability (Fig. S1 A). Optimization of the dilution factor should be performed on cells before applying it to the land-ExM procedure. At low concentrations, mCLING-atto647N can only stain the plasma membrane of the cells (Fig. S1, A and B). However, at high concentrations, mCLING-atto647N can stain the inner membrane structures of cells, such as mitochondria and the nuclear membrane (Fig. S1, C and D).

**land-ExM**

$0.0125 \times 10^6$ cells were seeded and cultured overnight on a plasma-cleaned coverslip attached to a custom PDMS chamber, 1 mm thick and 6.5 mm in diameter, as previously described (Shi et al., 2021). The chambers are included in the LR-ExM kit (cat#LR-RD488-050 or LR-MD568-050; Cytoseen) that we purchased, and they can also be made in a lab using the SYLGARD 184 Silicone Elastomer Kit. To label the whole lipids, cells were fixed with 37°C pre-warmed 3% paraformaldehyde and 0.1% glutaraldehyde in PBS for 10 min, followed by two washes with PBS and incubation with mCLING-Atto 647N (cat#710006AT1; Synaptic Systems) in PBS overnight at room temperature. Since mCLING may have batch-to-batch variation, concentration optimization (1:10 to 1:100 dilution in PBS) is recommended. After mCLING staining, the cells were fixed again with 37°C pre-warmed 3% PFA and 0.1% glutaraldehyde in PBS for 10 min, then proceeded to standard immunostaining steps if required. This included permeation with 0.1% Triton X-100 in PBS (PBST),

blocking with the blocking buffer (3% BSA in PBST), and primary antibody incubation overnight at 4°C. Secondary antibodies conjugated with LR-ExM DIG trifunctional probe NHS-Digoxigenin-MA (cat#LR-RD488-050 or cat#LR-MD568-050; Cytoseen) were used to fulfill multicolor land-ExM imaging. Incubate the cells with secondary antibodies (1:50 dilution in blocking buffer) for 1 h at room temperature.

To anchor and biotinylate whole proteins and anchor mCLING, incubate the cells with 2 mM NHS-Biotin-MA (cat#LD-P568-050 or #LD-P488-050; Cytoseen) in 100 mM NaHCO$_3$ for 1 h and wash with PBS three times. Cells are ready for gelation at this point. Cells were first incubated with pre-chilled monomer solution (8.6 g sodium acrylate, 2.5 g acrylamide, 0.15 g N,N′-methylenebisacrylamide, and 11.7 g sodium chloride in 94 ml PBS buffer) on ice for 5 min and then incubated with gelation solution (mixture of monomer solution, 10% [wt/vol] N,N,N′,N′-tetramethylethylenediamine stock solution, 10% [wt/vol] ammonium persulfate stock solution, and water at 47:1:1:1 volume ratio) on ice for 5 min. A coverslip was applied onto the top of the PDMS chamber to seal the cell-gelation solution to avoid oxygen interruption of the gelation procedure. The cell-gelation solution was then incubated at 37°C for 1–2 h. Gelled cells were immersed in heat denaturation buffer (200 mM sodium dodecyl sulfate, 200 mM NaCl, and 50 mM Tris, pH 6.8) for 1.5 h at 78°C and washed with excess of water for 30 min. To fluorescently label the whole proteins, immerse the gelled cells in poststaining buffer (10 mM HEPES and 150 mM NaCl, pH 7.5) twice, 30 min each time, then incubate the gelled cells with 4 µg/ml streptavidin dyes (included in cat#LD-P568-050/LD-P488-050; Cytoseen, or cat#016-540-084; Jackson ImmunoResearch) in poststaining buffer overnight. After 4 h of washing and expansion with an excess amount of DNase/RNase-free water, the gelled cells are ready for imaging.

To compare land-ExM with other protein contextual imaging methods, after permeation or antibody staining, cells were incubated in 25 mM NHS-MA (cat# 730300; Sigma-Aldrich) in 100 mM sodium bicarbonate for 1 h at room temperature, followed by three washes in PBS. After gelation and heat denaturation, gelled cells were washed with an excess amount of PBS, then incubated with 20 µg/ml NHS-Alexa Fluor 488 (cat#A20000; Invitrogen) in PBS or 1× SYPRO Orange (cat# S6651; Invitrogen) in PBS overnight. Details of reagents used can be found in the Key resources table in the Materials and methods section.

For users focusing on lipid context but with a low requirement for protein imaging, GMA (cat# 151238; Sigma-Aldrich) can serve as an alternative to NHS-biotin-MA (Fig. 1 A). Compared with the NHS-biotin-MA, using GMA as the anchor in the land-ExM workflow can result in equally high signal-to-noise ratio in lipid imaging but significantly dimmer signals in the protein channel. After permeation or antibody staining, cells were incubated with 0.04% (wt/vol) GMA in 100 mM sodium bicarbonate for 3 h at room temperature, followed by washing with PBS for three times. After gelation and heat denaturation, gelled cells were washed with an excess amount of PBS, then incubated with 20 µg/ml NHS-Alexa Fluor 488 (cat#A20000; Invitrogen) in PBS or 1× SYPRO Orange (cat# S6651; Invitrogen) in PBS overnight.

For users focusing on protein imaging only, the heat denaturation step can be replaced by proteinase K digestion. After gelation, the gelled cell was immersed overnight in digestion buffer (50 mM Tris, 1 mM EDTA, 0.5% [vol/vol] Triton X-100, and 1 M sodium chloride, pH 8) with freshly added 8 U/ml proteinase K (cat# P4850-5ML; Sigma-Aldrich), then washed with an excess amount of water for 30 min and an excess amount of poststaining buffer twice, 30 min each time. 4 µg/ml streptavidin dye in poststaining buffer was later added and incubated with the gelled cells overnight. After expansion with an excess amount of water, the gelled cell was ready for imaging.

### land-TREx
After mCLING and antibody staining as described in land-ExM, cells were incubated with 2 mM NHS-Biotin-MA (cat#LD-P568-050 or #LD-P488-050; Cytoseen) in 100 mM NaHCO$_3$ for 1 h and then washed three times with PBS. Cells were then incubated with pre-chilled TREx monomer solution (10.3 g sodium acrylate, 14.2 g acrylamide, and 0.09 g N,N′-methylenebisacrylamide in 95 ml PBS buffer) on ice for 5 min and gelation solution (a mixture of TREx monomer solution, 10% [wt/vol] N,N,N′,N′-tetramethylethylenediamine stock solution, 10% [wt/vol] ammonium persulfate stock solution, and water at 64:1:1:1 volume ratio) on ice for another 5 min. A coverslip was required to place onto the top of the PDMS chamber to seal the cell-gelation solution to avoid oxygen interruption of the gelation procedure. After 2-h gelation at 37°C in a humidity chamber, the gelled cells were immersed in heat denaturation buffer for 1.5 h at 78°C. The gelled cell was then washed with an excess amount of water for 30 min, an excess amount of poststaining buffer for 30 min, and an excess amount of poststaining buffer for another 30 min. 4 µg/ml streptavidin dye (cat#LD-P568-050 or #LD-P488-050; Cytoseen) in poststaining buffer was added later and incubated with gelled cells overnight. After another overnight gelled cell washing with an excess amount of water, the gelled cell was expanded around sevenfold and ready for imaging.

### land-pan-ExM
0.0125 × 10$^6$ cells were seeded and cultured overnight on a plasma-cleaned coverslip attached to a custom PDMS chamber (0.2 mm thickness and a 6.5 mm-diameter culture area). After fixation with 3% formaldehyde and 0.1% glutaraldehyde in PBS for 15 min at room temperature and rinsing with 1 × PBS once, cells were incubated with mCLING-Atto647N (cat#710006AT1; Synaptic Systems) overnight in PBS at room temperature. After mCLING staining, cells were fixed again with 3% formaldehyde and 0.1% glutaraldehyde in PBS for 15 min at room temperature. After washing with 1 × PBS twice, cells were incubated with 100 mM NaHCO$_3$ for 5 min, then incubated with 2 mM NHS-Biotin-MA (cat#LD-P568-050 or #LD-P488-050; Cytoseen) in 100 mM NaHCO$_3$ for 1 h at room temperature, and washed with PBS three times. Cells were then incubated with 0.7% formaldehyde and 1% (wt/vol) acrylamide in PBS at 37°C for 6–7 h, followed by PBS washing twice, 10 min each time. The cells were ready to its first round of gelation at this point.

To fulfill the first round of gelation, cells were first incubated with iExM first gelation solution (19% [wt/vol] sodium acrylate,

10% [wt/vol] acrylamide, 0.1% [wt/vol] N,N′-[1,2-Dihydroxy-ethylene]bisacrylamide, 0.25% [vol/vo] N,N,N′,N′-tetramethylethylenediamine, and 0.25% [wt/vol] ammonium persulfate in PBS) for 15 min at room temperature with a coverslip sealed at the top of the PDMS chamber, then moved to a humidity chamber and incubated for 1.5 h at 37°C. After gelation, the gelled cell was detached from the PDMS chamber and immersed in heat denaturation buffer for 15 min at room temperature, then incubated for 1 h at 78°C. Expand the gelled cells overnight with an excess amount of water. The gelled cell was ready for its second round of gelation at this step.

The expanded gelled cell was first incubated and gently shake with iExM second gelation solution (10% [wt/vol] acrylamide, 0.05% [wt/vol] N,N′-[1,2-Dihydroxyethylene]bisacrylamide, 0.05% [vol/vo] N,N,N′,N′-tetramethylethylenediamine, and 0.05% [wt/vol] ammonium persulfate in PBS) for three times, 20 min each time at room temperature, then sandwiched with two coverslips. To remove residual gelation solution, the sandwiched gelled cell was gently pressed with a Kimwipe inserted between the two coverslips. The sandwiched gelled cell was later incubated in a nitrogen-filled chamber at 37°C for 1.5 h. After detaching the two coverslips from the gelled cell, the gelled cell was incubated with 0.7% formaldehyde and 1% (wt/vol) acrylamide in PBS at 37°C for 6–9 h, followed by PBS washing three times, 30 min each time. The gelled cells were ready for their third round of gelation at this point.

The gelled cell was first incubated and gently shake with iExM third gelation solution (19% [wt/vol] sodium acrylate, 10% [wt/vol] acrylamide, 0.1% [wt/vol] N,N′-methylenebisacrylamide, 0.05% [vol/vo] N,N,N′,N′-tetramethylethylenediamine, and 0.05% [wt/vol] ammonium persulfate in PBS) for four times, 15 min each time, then sandwiched again with two coverslips. After the residual gelation solution was removed, the sandwiched gelled cell was incubated in a nitrogen-filled chamber at 37°C for 2 h. After detaching the two coverslips from the gelled cell, the gelled cell was incubated and shaken with 0.2 M NaOH for 1 h at room temperature, followed by PBS washing three times, 30 min each time. The gelled cell was then incubated overnight with 4 µg/ml streptavidin dye (cat#LD-P568-050 or #LD-P488-050; Cytoseen) in poststaining buffer. After overnight expansion with an excess amount of water, the cell-gel was expanded to around 12-fold and was ready for imaging.

## Imaging
Gelled cells imaging was all performed on an Airyscan confocal microscope (ZEISS LSM 980 with Airyscan 2) with a 63× water-immersion objective (Zeiss LD C-Apochromat 63×/1.2 W Corr M27) with effective lateral resolution at 138 nm (measured by TetraSpeck Microspheres, 0.1 µm, fluorescent blue/green/orange/dark red). The effective resolution of extended samples is calculated as this microscope resolution divided by the expansion factor, ranging from 3.7 to 12.0. The highest effective resolution we could achieve was 12 nm. Airyscan SR4Y—best signal mode with a 0.2 AU pinhole and 1.25 AU total detection area—was used for the 3D imaging of all the samples.

## Image rendering
Images in Fig. 4, E, J, K, P, and Q were processed with a custom-written MATLAB script. Briefly, stack images were denoised by a Wiener filter, smoothed by a Gaussian filter, and then binarized. Binarized stack images were reconstructed in Imaris 10.2 using the pre-expansion dimensions. Surfaces were created, and transparency was adjusted for visualization of inner structures. The MATLAB codes are available at Xiaoyu Shi Lab's GitHub page: https://github.com/XYShi-Lab/land-ExM. All the other images were simply contrasted or projected using basic functions of Fiji (ImageJ).

## Quantification and statistical analysis
All images were processed and analyzed using ImageJ and Custom MATLAB code. All graphs were generated using Prism 10 (GraphPad software).

## Reagent availability
All reagents used in this work are commercially available, including the land-ExM protein label, NHS-biotin-MA, and LR-ExM trifunctional antibodies. See Key resources table.

## Online supplemental material
Fig. S1 to Fig. S7 can be found in the supplemental information. Fig. S1 shows mCLING optimization for lipid staining of cells. Fig. S2 shows alternative land-ExM workflow to avoid cross talk between NHS-biotin-MA and mCLING. Fig. S3 shows land-ExM using proteinase K digestion. Fig. S4 shows land-ExM coupled with immunostaining LR-ExM for lipid vesicle identification. Fig. S5 shows land-ExM coupled with immunostaining LR-ExM for membrane-bound organelle visualization. Fig. S6 shows land-ExM coupled with immunostaining for cytoskeleton visualization. Fig. S7 shows land-ExM reveals SGs at different locations of cells.

## Data availability
The MATLAB codes for 3D rendering are available at Xiaoyu Shi Lab's GitHub page: https://github.com/XYShi-Lab/land-ExM. All the other data analyses were performed using the open-source software Fiji (ImageJ) and are available in the published article and online supplemental material.

## Acknowledgments
We thank Joerg Bewersdorf's lab for sharing the pan-ExM protocol and Paul Tillberg's lab for consulting on TREx. We also thank Dr. Kiryl Piatkevich for recommending SYPRO Orange dye.

Y. Zhuang and X. Shi are supported by the National Institutes of Health Director's New Innovator Award (DP2GM150017, 3DP2GM150017-01S1) and the NSF Faculty Early Career Development Program (CAREER) Award (2341058). Z. Zhang and Z. Dai are supported by the Chan Zuckerberg Initiative Advancing Imaging Through Collaborative Projects Award.

Author contributions: Yinyin Zhuang: conceptualization, data curation, formal analysis, investigation, methodology, project administration, validation, visualization, and writing—original draft, review, and editing. Zhao Zhang: methodology. Zhipeng

Dai: methodology and resources. Xiaoyu Shi: conceptualization, data curation, funding acquisition, methodology, project administration, resources, supervision, validation, and writing—original draft, review, and editing.

Disclosures: All authors have completed and submitted the ICMJE Form for Disclosure of Potential Conflicts of Interest. X. Shi reported other from Cytoseen Inc. outside the submitted work. No other disclosures were reported.

Submitted: 7 February 2025

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

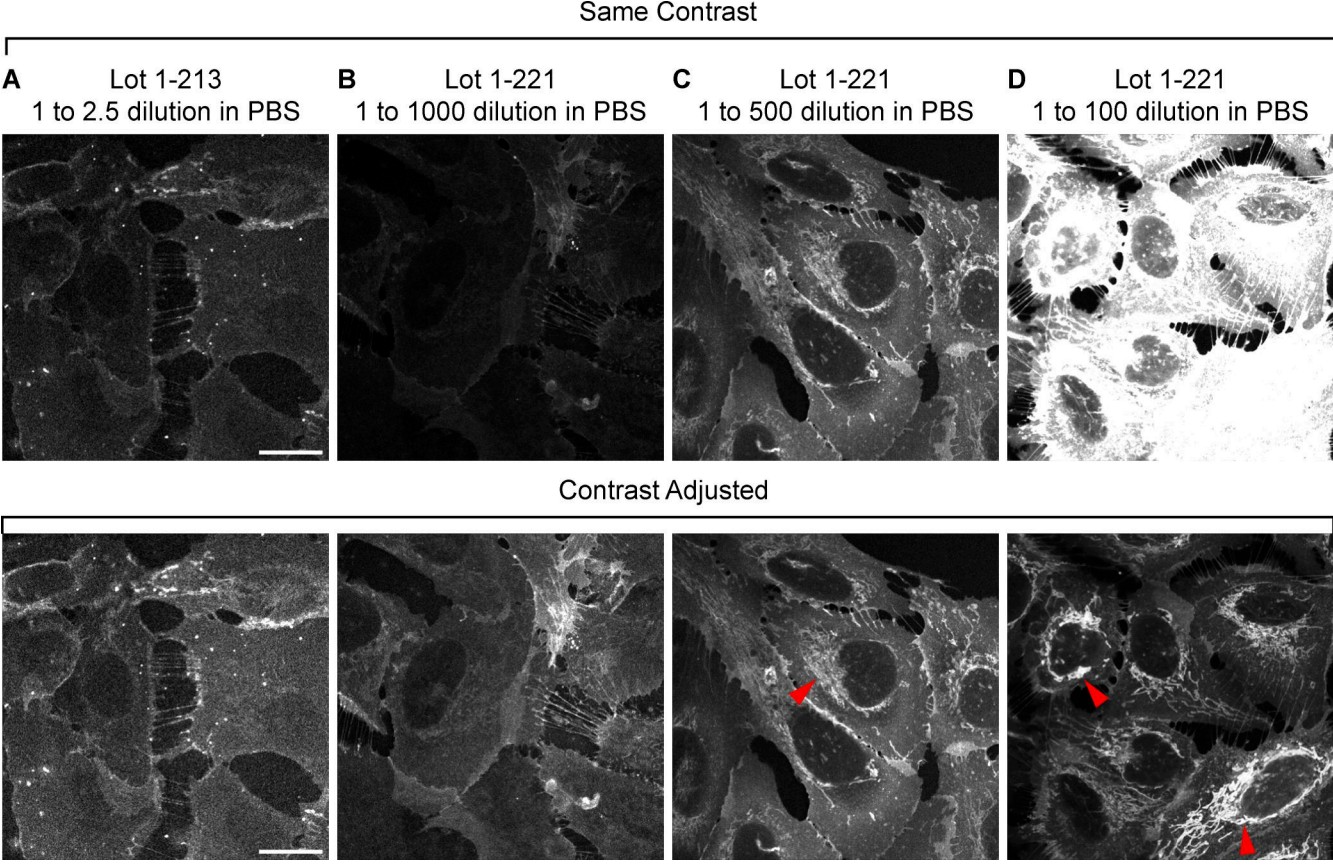

Figure S1.   **mCLING optimization for lipid staining of cells. (A–D)** Airyscan images of U2OS cells stained with different batches of mCLING at different dilution factors. Scale bars: 20 µm. Red arrowheads indicate lipid structures in the cytoplasm. All images were taken with an Airyscan microscope.

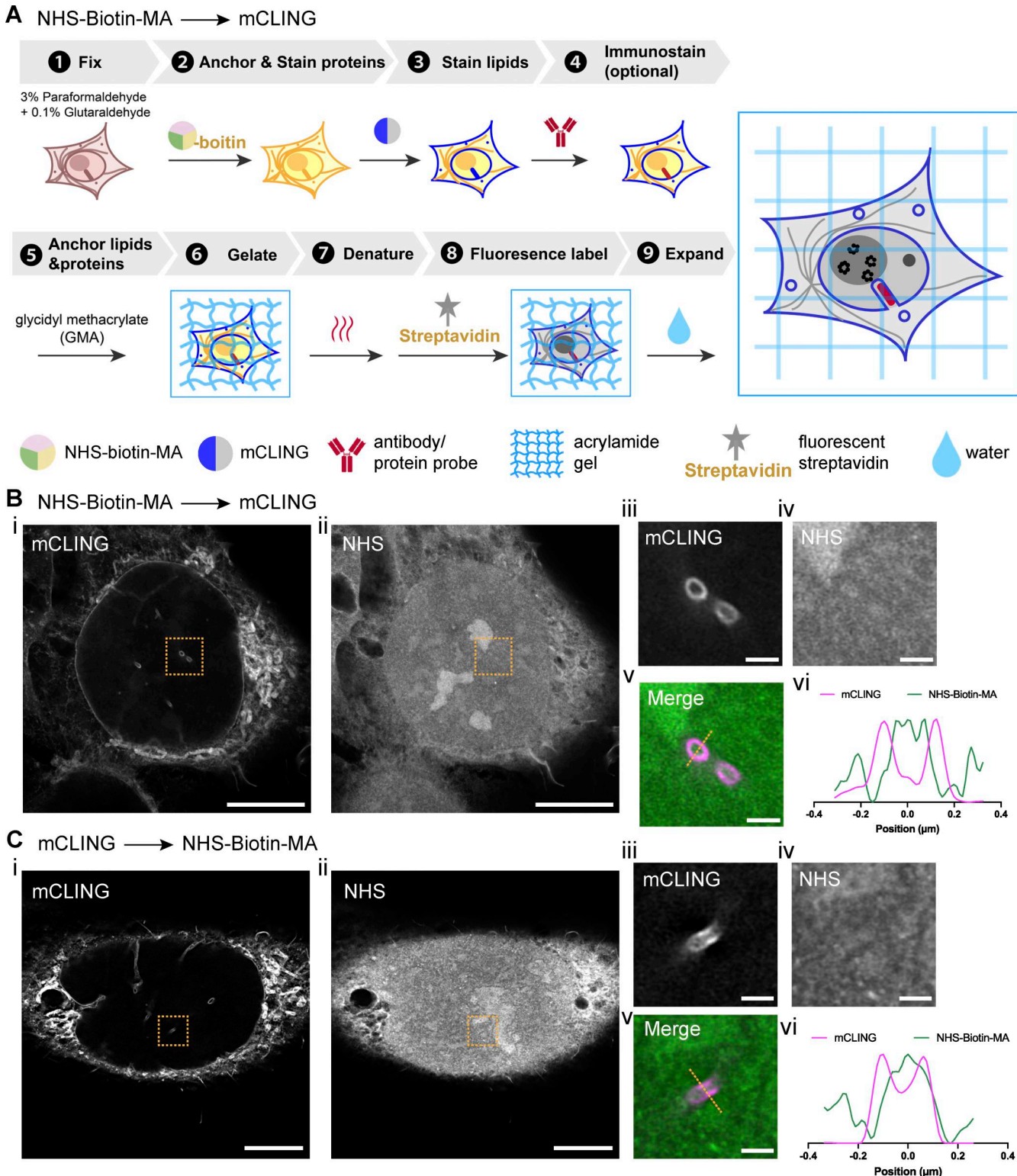

Figure S2. **Alternative land-ExM workflow to avoid cross talk between NHS-biotin-MA and mCLING. (A)** Alternative workflow of land-ExM. **(B)** i and ii: land-ExM images of U2OS cells stained first with NHS-biotin-MA and then mCLING. iii to v: Magnified images of boxes in i and ii. vi: Normalized intensity profile along the orange line in v. Scale bar: 5 μm in pre-expansion unit. Linear expansion factor: 4.0 (i and ii). 0.5 μm in pre-expansion unit. Linear expansion factor: 4.0 (iii to v). **(C)** i and ii: land-ExM images of U2OS cells stained first with mCLING and then NHS-biotin-MA. iii to v: Magnified images of orange boxes in i and ii. vi: Normalized intensity profile along the orange line in v. Scale bar: 5 μm in pre-expansion unit. Linear expansion factor: 4 (i and ii). 0.5 μm in pre-expansion unit. Linear expansion factor: 4.0 (iii to v). All images were taken with an Airyscan microscope.

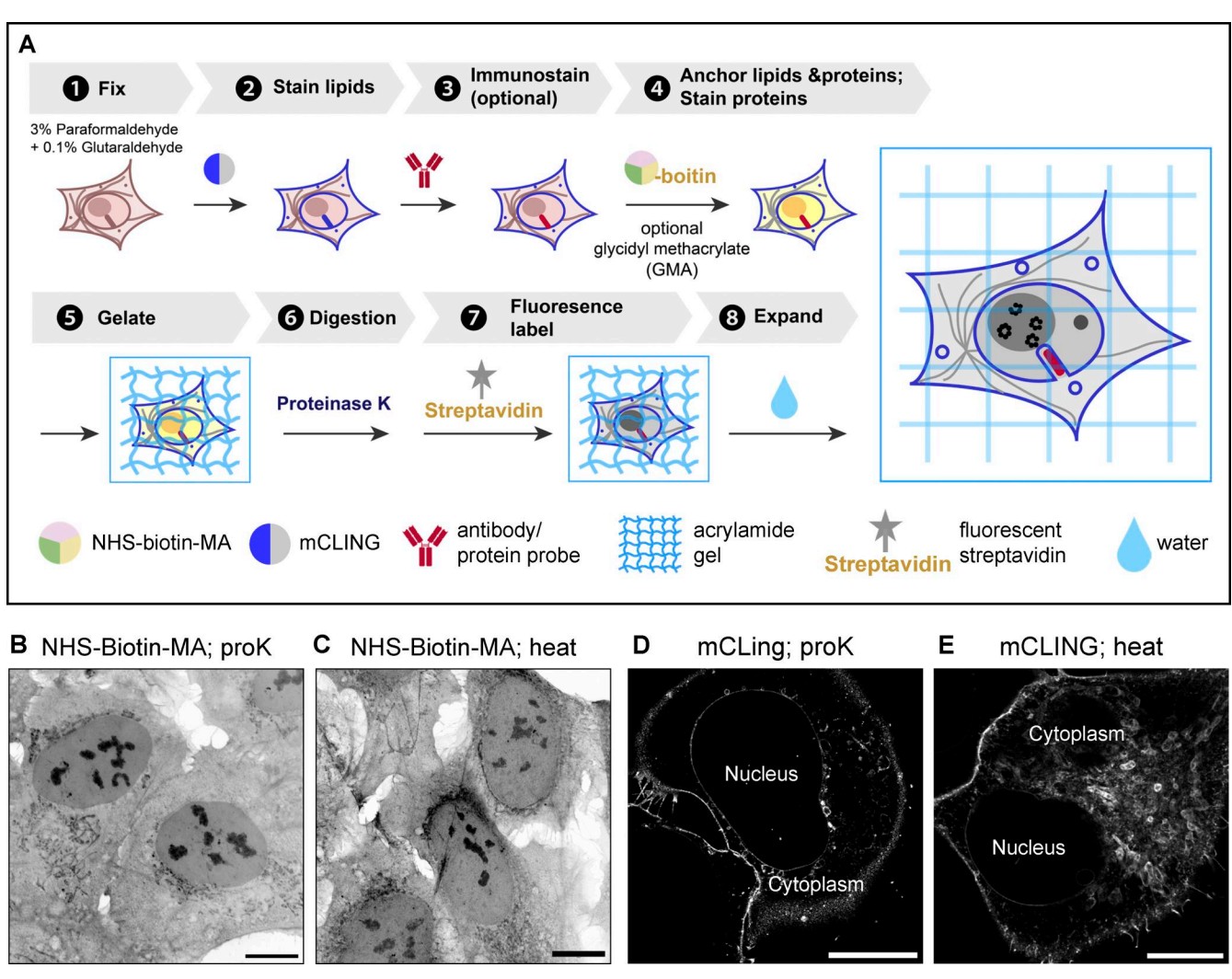

Figure S3. **land-ExM using proteinase K digestion. (A)** Workflow of land-ExM using proteinase K digestion to homogenize cells instead of heat denaturation. **(B)** land-ExM protein image of U2OS cells with proteinase K digestion (proK). Scale bar: 10 µm in pre-expansion unit. Linear expansion factor: 4.0. **(C)** land-ExM protein image of U2OS cells with heat denaturation (heat). Scale bar: 10 µm in pre-expansion unit. Linear expansion factor: 4.0. **(D)** land-ExM lipid image of U2OS cell with proteinase K digestion (proK). Scale bar: 10 µm in pre-expansion unit. Linear expansion factor: 4.0. **(E)** land-ExM lipid image of U2OS cells with heat denaturation (heat). Scale bar: 10 µm in pre-expansion unit. Linear expansion factor: 4.0. All images were taken with an Airyscan microscope. Image in C is also shown in Fig. 1 D.

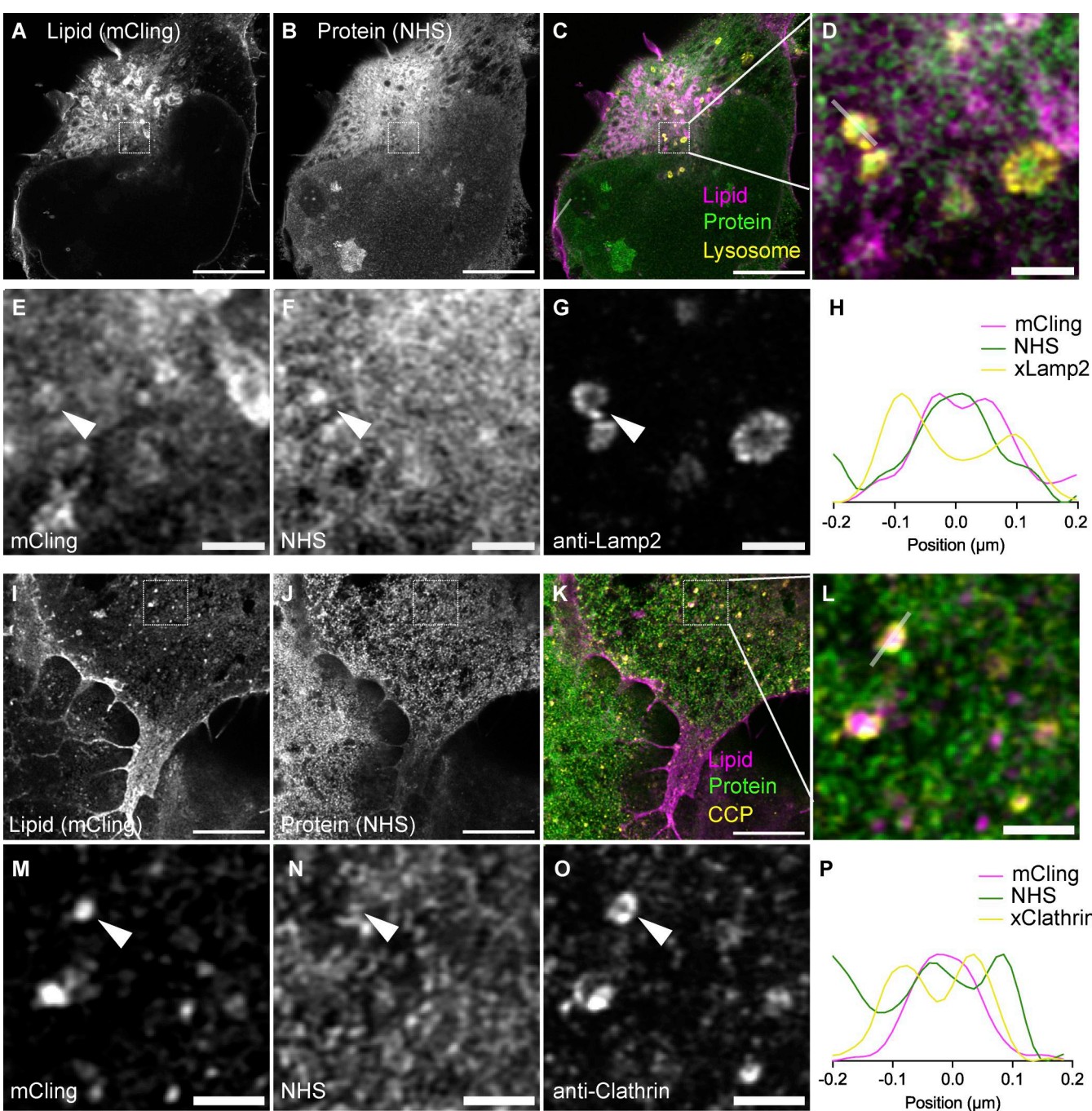

Figure S4. **land-ExM coupled with immunostaining LR-ExM for lipid vesicle identification. (A–C)** land-ExM lipid (magenta) and protein (green) images of U2OS cells immunostained with anti-Lamp2 antibodies (yellow). The anti-Lamp2 antibodies are labeled LR-ExM second antibodies, which are second antibodies conjugated with NHS-digoxigenin-MA. Scale bar: 5 μm in pre-expansion unit. Linear expansion factor: 4. **(D–G)** Magnified images of A–C showing details of lysosomes. Scale bar: 500 nm in pre-expansion unit. **(H)** Intensity profile along the gray line across the lysosome in image (D). **(I–K)** land-ExM lipid (magenta) and protein (green) images of U2OS cells immunostained with anti-clathrin antibodies (yellow). The anti-clathrin antibodies are labeled LR-ExM second antibodies, which are second antibodies conjugated with NHS-digoxigenin-MA. Scale bar: 5 μm in pre-expansion unit. Linear expansion factor: 4. **(L–O)** Magnified images of I–K showing details of clathrin-coated pits. Scale bar: 500 nm in pre-expansion unit. **(P)** Intensity profile along the gray line across the clathrin-coated pit in image (L). All images were taken with an Airyscan microscope.

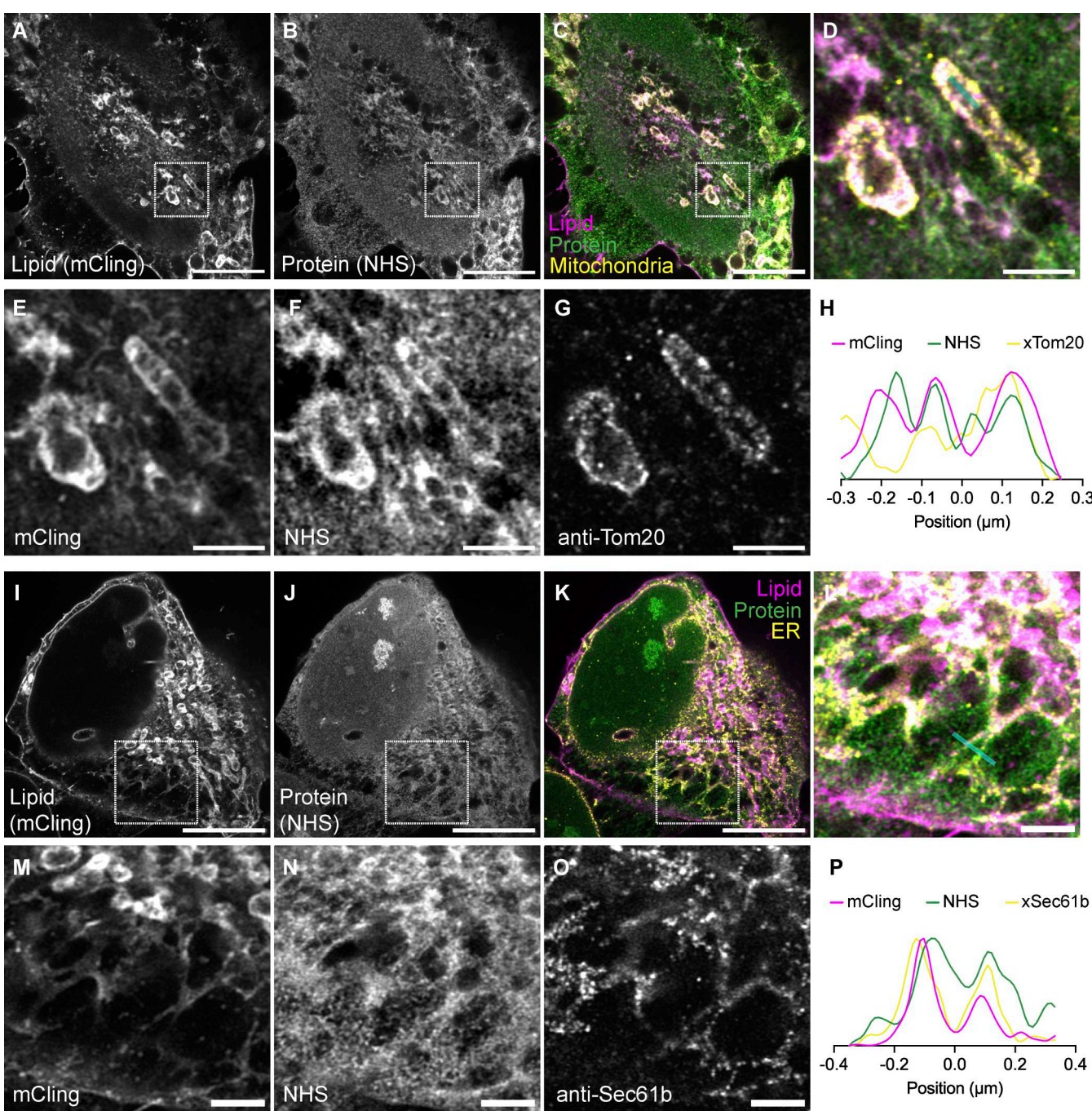

Figure S5. **land-ExM coupled with immunostaining LR-ExM for membrane-bound organelle visualization. (A–C)** land-ExM total lipid (magenta) and protein (green) images of U2OS cells immunostained with anti-Tom20 antibodies (yellow). The anti-Tom20 antibodies are labeled LR-ExM second antibodies, which are second antibodies conjugated with NHS-digoxigenin-MA. Scale bar: 5 μm in pre-expansion unit. Linear expansion factor: 4. **(D–G)** Magnified images of A–C showing details of mitochondria. Scale bar: 1 μm in pre-expansion unit. **(H)** Intensity profile along the cyan line across the mitochondria in image (D). **(I–K)** land-ExM lipid (magenta) and protein (green) images of U2OS cells immunostained with anti-Sec61b antibodies (yellow). The anti-Sec61b antibodies are labeled LR-ExM second antibodies, which are second antibodies conjugated with NHS-digoxigenin-MA. Scale bar: 5 μm in pre-expansion unit. Linear expansion factor: 4. **(L–O)** Magnified in images of I–K showing details of ER. Scale bar: 1 μm in pre-expansion unit. **(P)** Intensity profile along the cyan line across the ER in image (L). All images were taken with an Airyscan microscope.

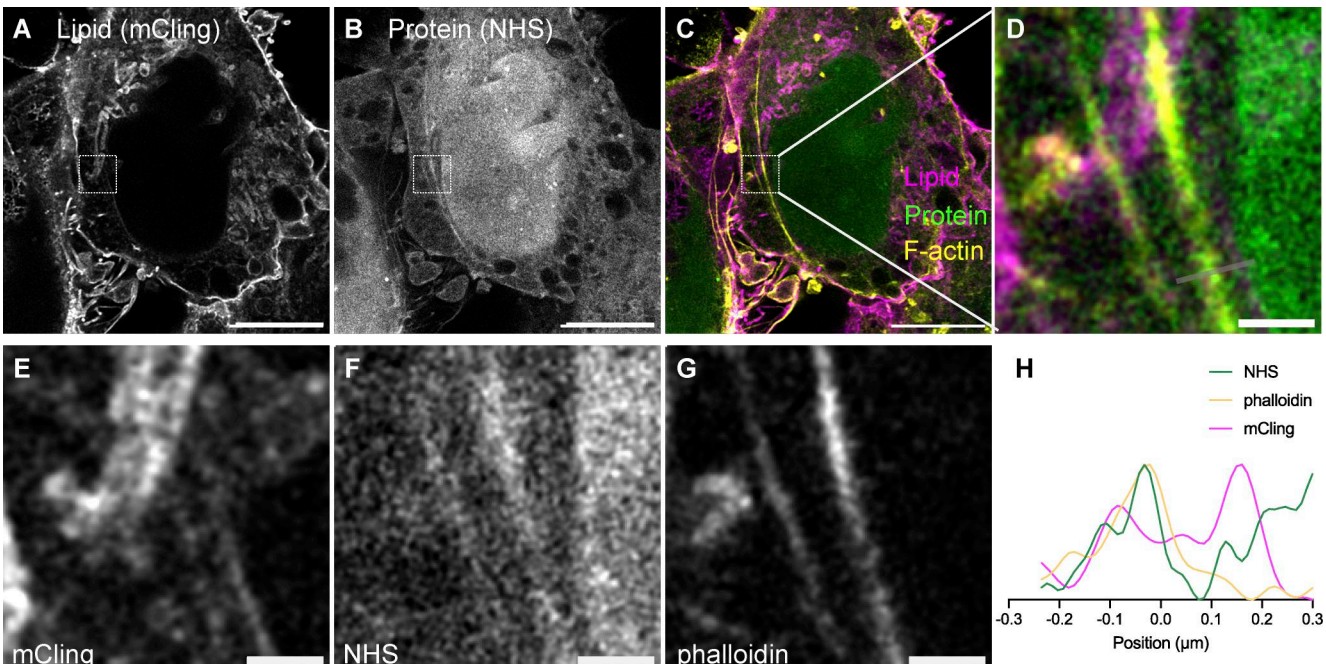

Figure S6. **land-ExM coupled with immunostaining for cytoskeleton visualization. (A–C)** land-ExM total lipid (magenta) and protein (green) images of breast cancer cells stained with phalloidin-FITC and anti-FITC antibody (yellow). Scale bar: 5 μm in pre-expansion unit. Linear expansion factor: 3.8. **(D–G)** Zoom in images of A–C showing details of F-actin. Scale bar: 500 nm in pre-expansion unit. **(H)** Intensity profile along the gray line in image (D). All images were taken with an Airyscan microscope.

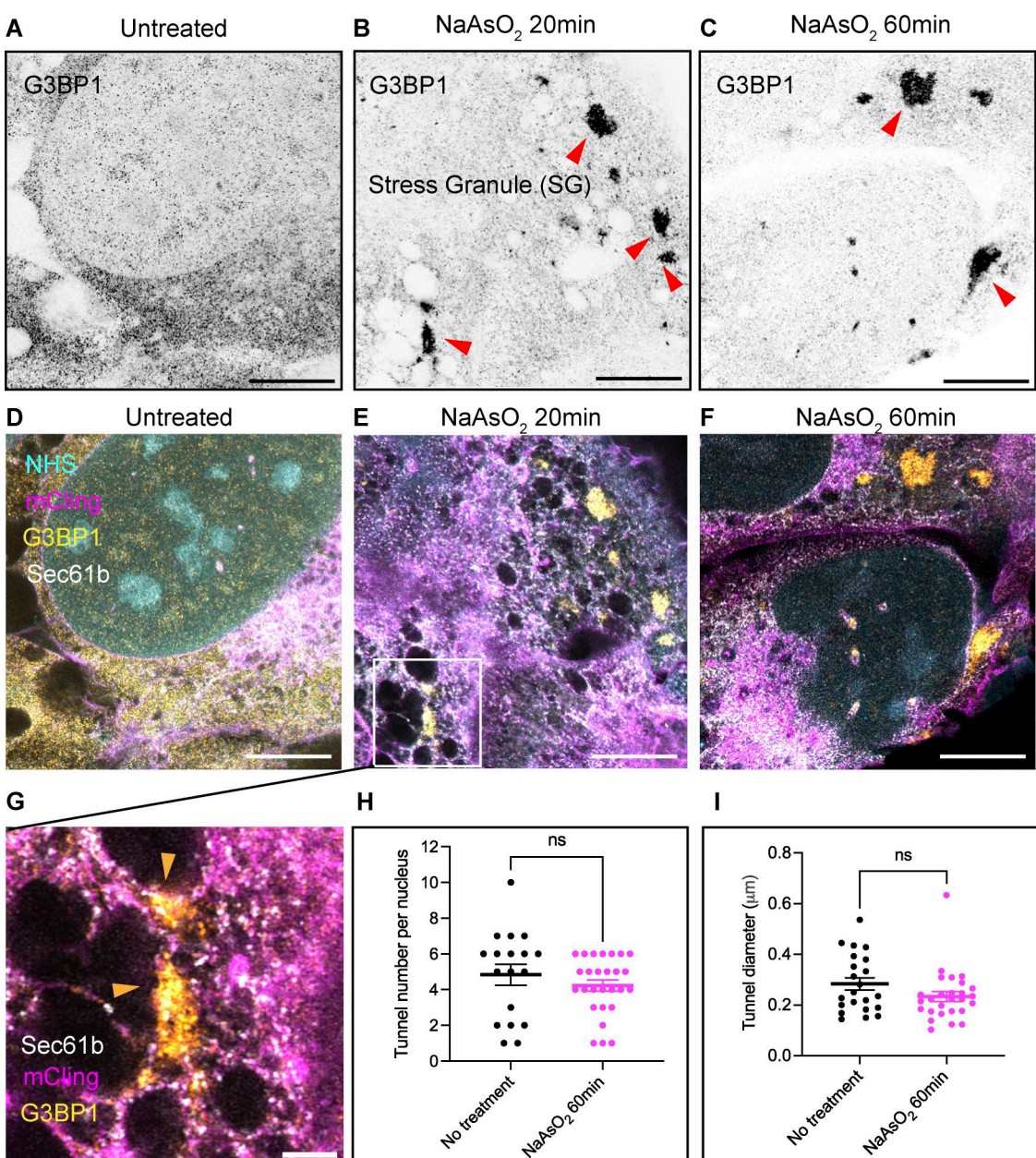

Figure S7. **land-ExM reveals SGs at different locations of cells. (A–C)** land-ExM images of U2OS cells untreated or treated with NaAsO2 for 20 or 60 min, then immunostained with anti-G3BP1 antibody. Scale bar: 5 μm in pre-expansion unit. Linear expansion factor: 4. **(D–F)** land-ExM images of U2OS cells stained with mCLING (magenta) and NHS ester dye (cyan) and immunostained with anti-G3BP1 (yellow) and anti-Sec61b (white) antibodies. Cells were untreated or treated with NaAsO2 for 20 min or 60 min. Scale bar: 5 μm in pre-expansion unit. Linear expansion factor: 4. **(G)** Magnified images of E showing SGs formed adjacent to ER (orange arrowheads). Scale bar: 1 μm in pre-expansion unit. **(H)** Analysis of the number of nuclear tunnels per cell with or without 60 min NaAsO2 treatment. Each bar represents the mean ± standard error of more than 18 cells. The ns indicates P > 0.05 by Welch's *t* test. **(I)** Analysis of the diameter of nuclear tunnels in cells with or without 60 min NaAsO2 treatment. Each bar represents the mean ± standard error of more than 20 cells. ns indicates P > 0.05 by Welch's *t* test. All images were taken with an Airyscan microscope. The cell shown in F is also shown in Fig. 4, A, F, and L.

