## [Peer Review File · The Journal of Cell Biology]

Landscape Expansion Microscopy Reveals Interactions between Membrane and Phase-Separated Organelles

Yinyin Zhuang, Zhao Zhang, Zhipeng Dai, and Xiaoyu Shi

Corresponding Author(s): Xiaoyu Shi, University of California, Irvine

Review Timeline:

Submission Date:	2025-02-07
Editorial Decision:	2025-04-07
Revision Received:	2025-09-22
Editorial Decision:	2025-11-30
Revision Received:	2025-12-07

Monitoring Editor: Joerg Bewersdorf

Scientific Editor: Dan Simon

Transaction Report:

DOI: <https://doi.org/10.1083/jcb.202502035>

April 7, 2025

Re: JCB manuscript #202502035

Xiaoyu Shi
University of California, Irvine

Dear Dr. Shi,

Thank you for submitting your manuscript entitled "Landscape Expansion Microscopy Reveals Interactions between Membrane and Phase-Separated organelles." The manuscript has been evaluated by expert reviewers, whose reports are appended below. Unfortunately, after an assessment of the reviewer feedback, our editorial decision is against publication in JCB.

You will see that while the Reviewers feel that land-ExM could be a promising method they also raise substantial concerns about lack of sufficient comparison with existing expansion microscopy methods and a clear demonstration that land-ExM provides an advantage and reveals structures that existing methods do not. The Reviewers also pointed out an absence of quantitative analysis of the observed contact sites and condensates as well as limited novelty provided by the imaging of nuclear invaginations.

While your manuscript is intriguing, we feel that addressing the reviewer comments would require a more substantial amount of new experiments than can be done in a typical revision period. If you wish to expedite publication of the current data, it may be best to pursue publication at another journal.

However, given interest in the topic, we would be open to resubmission to JCB of a significantly revised and extended manuscript. A revised paper would most likely be best suited for the JCB Tools format. This would require refocusing the work to make it clear that the key advance is the adaptation of existing anchors to improve simultaneous imaging of proteins and lipids and provide comparisons with existing methods. Additionally, a more quantitative analysis of the nuclear invaginations and their potential contacts or proximity with other organelles and condensates as well as showing that land-ExM is a better option for imaging of these types of structures would be necessary.

Alternatively, you could keep this as an Article but that would require a much more in-depth analysis of the nuclear invaginations and detailed characterization of how they make contacts with other organelles and condensate structures. The bar for biological novelty would be much higher for an Article but for this format it would not be necessary to compare land-ExM with other methods.

If you would like to resubmit this work to JCB, please contact the journal office to discuss an appeal of this decision or you may submit an appeal consisting of a detailed revision plan with a point-by-point response directly through our manuscript submission system. Please note that priority and novelty would be reassessed at resubmission of a revised manuscript.

Regardless of how you choose to proceed, we hope that the comments below will prove constructive as your work progresses. We would be happy to discuss the reviewer comments further once you've had a chance to consider the points raised in this letter. You can contact the journal office with any questions at cellbio@rockefeller.edu.

Thank you for thinking of JCB as an appropriate place to publish your work.

Sincerely,

Joerg Bewersdorf, PhD
Monitoring Editor
Journal of Cell Biology

Dan Simon, PhD
Scientific Editor
Journal of Cell Biology

Reviewer #1 (Comments to the Authors (Required)):

This manuscript introduces Landscape Expansion Microscopy (Land-ExM), which incorporates a trifunctional anchoring strategy (NHS-biotin-MA) to improve biomolecule retention in expansion microscopy (ExM), particularly for proteins and lipids. The

authors claim that this anchoring strategy enhances fluorescence signal intensity and enables high-resolution visualization of membrane contact sites (MCSs) and phase-separated organelles, achieving an effective resolution of ~30 nm with Airyscan microscopy. They demonstrate Land-ExM through imaging quadruple-organelle contact sites and mitochondrial-nuclear interactions, which they suggest are challenging to observe using conventional ExM techniques.

While the proposed anchoring strategy improves fluorescence intensity, the manuscript does not convincingly establish Land-ExM as a significant advancement over existing ExM methods. The resolution claim is misleading because the 30 nm resolution is due to the combination of 4× Expansion and Airyscan microscopy rather than the expansion process itself, meaning the intrinsic resolution remains ~60-70 nm using diffraction-limited microscopes, similar to conventional 4× ExM. There is no direct benchmarking against other high-resolution ExM techniques, such as TReX, iExM, Pan-ExM, or Expansion Revealing, which already achieve comparable or superior results through different expansion factors or anchoring strategies. Furthermore, the demonstration experiments fail to establish that the observed organelle interactions are uniquely resolvable by Land-ExM, as the previously mentioned ExM methods combined with mCLING pre-staining could likely yield the same or better results.

Additionally, NHS-biotin-MA should theoretically have similar immunofluorescence signal retention as MA-NHS anchoring method, which has been reported by Chozinski et al. in 2016, making it unclear whether Land-ExM provides any real advantage for antibody-based labeling. Given these issues, the manuscript requires major revisions to properly validate its claims through comparative studies, clarify its resolution improvements, and provide evidence that it offers distinct advantages over existing techniques. However, even with these improvements, the contribution may be more appropriate for a specialized microscopy or bioimaging journal rather than a broad high-impact venue.

Comments in details:

1. The resolution claim of ~30 nm is not due to Land-ExM's expansion process itself but rather largely to the Airyscan microscopy, which technically is a super-resolution microscope at its own right. Since the expansion factor remains at 4×, the actual improvement in spatial resolution is limited to ~60-70 nm under diffraction-limit microscopes, similar to other 4× ExM methods. The authors should clarify that Land-ExM does not inherently improve spatial resolution beyond conventional 4× ExM and avoid misleading comparisons with high-expansion techniques like 10× ExM or iExM.
2. The manuscript lacks a rigorous comparison with other ExM methods, making it difficult to assess whether Land-ExM provides a meaningful advantage. The only quantitative comparison focuses on fluorescence intensity using SYPRO Orange and NHS-Alexa Fluor 488, but these reagents were not used in a way that aligns with standard ExM workflows. Typically, SYPRO Orange and NHS-Alexa Fluor 488 are applied post-expansion, not before expansion, to avoid fluorophore degradation during the gelling process. Additionally, SYPRO Orange does not covalently bind to proteins, meaning it is largely washed away during homogenization, which explains its weak signal in the figure. However, this weak signal is misleading, as it does not accurately reflect the retention capabilities of conventional ExM approaches. Neither SYPRO Orange nor NHS-Alexa Fluor 488 are anchoring reagents and thus comparing them directly with Land-ExM's trifunctional anchor does not provide meaningful insight into biomolecule retention. A proper comparison would use MA-NHS as the anchoring reagent for the control group (PMID: 27064647) and then apply SYPRO Orange and NHS-Alexa Fluor 488 post-expansion, which would allow for an accurate assessment of Land-ExM's effectiveness. Without this, it remains unclear whether Land-ExM truly improves biomolecule retention beyond what existing methods offer.
3. The manuscript does not provide data comparing immunofluorescence (IF) efficiency, making it unclear whether Land-ExM offers any real advantage in retaining antibody-based labels. In principle, this anchoring approach is not expected to improve IF retention compared to MA-NHS anchoring, as the biotin-streptavidin amplification strategy does not enhance antibody labeling, which already relies on covalent binding or strong antigen-antibody interactions. Additionally, MA-NHS anchoring method has been reported by Chozinski et al. in 2016, which further decreases the novelty of this work. Without direct comparisons of pre-expansion vs. post-expansion immunofluorescence signal intensity between Land-ExM and a standard NHS-MA anchoring protocol, there is no evidence that Land-ExM improves IF-based ExM workflows. Similarly, the study lacks a quantitative comparison of lipid probe retention efficiency, particularly for mCLING, which is a key marker for membrane structures. Since mCLING is already biotinylated, the use of streptavidin in Land-ExM could simply enhance its signal rather than genuinely improving lipid retention, making it unclear whether this anchoring approach provides any retention benefits over standard lipid-labeling strategies. To properly evaluate Land-ExM's performance, the authors should provide quantitative retention efficiency comparisons for both immunofluorescence signals and lipid probes (e.g., mCLING) against conventional ExM anchoring methods.
4. The claim that Land-ExM uniquely enables the visualization of mitochondrial-nuclear invagination contacts is not supported. Other ExM methods, particularly those with higher expansion factors (10× ExM, iExM, Pan-ExM, Expansion Revealing), could likely resolve these structures if the sample was pre-stained with mCLING. The authors do not provide evidence that these contacts could not have been observed with alternative ExM approaches. To support their claim, they should either compare Land-ExM to another ExM method for MCS imaging or revise their statement to acknowledge that other methods could achieve similar results.
5. The streptavidin-biotin labeling strategy introduces potential artifacts, as fluorophore-bearing streptavidin binds to both biotinylated proteins and biotinylated lipid probes (e.g., mCLING), which could cause non-specific signal overlap.

Reviewer #2 (Comments to the Authors (Required)):

Zhuang et al., in their manuscript "Landscape Expansion Microscopy Reveals Interactions between Membrane and Phase-Separated organelles" describe a new microscopy technique, Landscape Expansion Microscopy (land-ExM) and demonstrate its

application to multi-organelle contacts. Overall it is strong work and I recommend publication after addressing the minor points below.

Land-ExM is based on earlier work by Prof. Shi on label-retention expansion microscopy (LR-ExM) that uses trifunctional small-molecule probes containing an NHS ester group to label proteins nonspecifically together with the mCLING probe to label lipids. These can be combined with immunostaining to enable a number of novel experiments. Similar experiments were reported by Damstra et al., *eLife* 2022 (cited ref. 19). In those experiments, Damstra 2022 performed similar measurements using NHS ester-functionalized dyes to label proteins and mCLING to label lipids, however the authors of that paper commented that the protein and lipid signals were very similar to one another. In contrast, Zhuang et al. now report and improve protocol that achieves distinct lipid and protein signals, each of which is highly information rich. It is a valuable contribution with high-quality results and it will be of interest to a broad audience.

I have many minor points that can be addressed with simple revisions to the text or figures. In one case, a small measurement may be helpful, but I do not see this as required.

- Many acronyms are indicated in Fig. 1a. These should ALL be spelled out for clarity to the readers in the figure or caption, including PFA, GA, IF, STV. In addition, on panel A) step 5 the spelling should be "gelate".
- Fig 1H contains beautiful data. I suggest that the authors show the positions where the different views correspond using, for instance, some dashed lines. This helps the reader gain a better appreciation of the 3D structure within their data.
- I think it has become a bit fashionable to display pan-ExM or similar NHS labeling type images in electron microscopy (EM) style with black on white to make a point about how that staining is conducted with low molecular specificity similar to EM. This is a purely stylistic point, but I'd encourage authors to think hard about what format is generally easiest for the reader to observe for the type of data overall and to stick with that (I suspect it is white on black). For instance, multichannel data with blue lipid and black protein signal in Fig. 1 don't particularly go well together since the overlay doesn't change the hue...for this reason, typically fluorescence uses magenta/green for two-channel data because a distinctive white overlay is easier to perceive. The authors later switch to bright on dark in Figures 3, S2, and S3 which I personally prefer. Figure 3 does put the protein signal as a white on black as I suggest, but white doesn't go super well with red. Why not just stick with standard 2- and 3-channel color schemes like MG or RGB or CMY?
- While I'm on the subject of figure format, I'll just mention that many people consider it a best practice when displaying display separate channels (from a multicolor image) as grayscale images not color images. This avoids issues related to different levels of color perception. For instance see Jayme Johnson, MBoC, 2012 or the MBoC color guide <https://www.ascb.org/diversity-equity-and-inclusion/how-to-make-scientific-figures-accessible-to-readers-with-color-blindness/>. This is relevant for Figures S2 & S3.
- Figure 2 panel N would be improved by including a legend for the color bar to know what the heights are for the color bar (e.g., blue = 0um and white =6um or something).
- The caption and text of Fig 2 is not clear regarding what signal is in what channel. The caption indicates that Land-ExM reveals things but as I understand it Land-ExM is a dual protein and lipid stain. I get the impression that Fig 2a-g shows the protein signal because it is black and white like what we saw in Fig 1, and then maybe H-M are lipid signal but this would be important to state in the figure and/or caption.
- Line 57 of the supplement should probably be for (P) not (H) in Fig S2. Same issue for the Fig. S3 caption.
- The authors compare protein signal for their method and for cells treated first with MA-NHS and then At488-NHS, showing that their method produces ~10x more signal. One might expect a two-step procedure might be more sensible with 1mM dye-NHS stain first followed by the 25 mM MA-NHS step.
- Can the authors first do the MA-biotin-NHS step and then the mCLING step (with another GMA or MA-NHS step at the end to retain mCLING if necessary) to avoid any potential crosstalk? The authors could address this with a comment around line 162 where the possibility of cross-contamination is described, although it may be helpful to perform a test of this (or to show one in the supplementary information, if they have already performed such a test).
- A spatial resolution of ~30nm is asserted. While some line profiles are provided in Figures S2 & S3 the 30nm spatial resolution is not evident or commented on. This is not a major point. I suggest that the authors either show some evidence for 30 nm resolution or else simply suggest that a spatial resolution of 30 nm could be achieved with the method.
- On line 128 the authors estimate stoichiometry of dyes per NHS-biotin-MA molecule under the assumption that one streptavidin binds one NHS-biotin-MA group, however, streptavidin could bind multiple biotins if they were in sufficient proximity, albeit with different affinity.

- On line 132 the authors state that their method has significantly higher signal-to-noise ratio compared to ExM with Alexa Fluor 488. The authors showed significantly higher SIGNAL for their method (which is great) but they do not show any measurements of signal/noise. This could be fixed with a small text revision.
 - I'm confused around line 265 why mitochondria are re-visited and sort of validated when that seemed to have already been done around line 178. Does it make sense to do this twice?
 - Around line 330, the authors might clarify slightly that they mean "Utilizing land-ExM labeling together with super-resolution techniques...", if this is indeed the case.
 - Line 53 of supplement should say "Linear expansion."
 - The authors may wish to also cite the 2019 lipid expansion microscopy bioRxiv manuscript from the Boyden group at the same place where they cite the 2024 bioRxiv (cited reference 33).
 - The authors show in Fig. 3 a fascinating observation with nuclear invaginations and multiple organelle contacts. I see this paper as primarily a method paper that also demonstrates utility on an application, but that the number of cells and such that are analyzed for the various organelle contacts is small (1-2 cells?) so I wonder whether it would be more appropriate to de-emphasize the biological conclusions and to make it clear that the new tool facilitates observations.
- For instance, on line 220, the authors indicate "This spatial relationship was commonly seen in nuclear invaginations of cells under stress in our experiments." Similarly, on line 229 the authors indicate "ER-SG contacts inside nuclear tunnels were frequently observed in cells under stress in our experiments." But in each case, only one cell was shown or apparently scored for these phenotypes.
- I would suggest the authors determine whether they want to change the tone and scope to say that these features can be easily observed using this new method, or if they want to present some statistics or quantification on the phenotype such as how often it is observed in the treated vs control cells.

Reviewer #3 (Comments to the Authors (Required)):

Summary: In this manuscript the authors describe an approach that they term landscape expansion microscopy or "land-ExM", which uses trifunctional labels that self-anchor to the gel that allows for 4-fold expansion upon hydration, which they combine with super-resolution microscopy. Land-ExM is an extension of their previously published approach termed label-retention (LR) ExM - the major new aspect of this new work is the addition of lipid labeling using the mCLING reagent. Applying the approach to U2OS cells, the authors report several observations including tunnels through the nucleus that are frequently near nucleoli, that contain labeling for Sec61, and that have stress granule components within them in response to stress and are often found in proximity to mitochondria at the nuclear surface.

Overall assessment: Technically the land-ExM approach appears to be capable of combining total protein and lipid information and is compatible with immunofluorescence microscopy. Membrane labeling has remained a major challenge in expansion microscopy until the past few years and represents an important aspect of future developments. How well land-ExM performs compared to other approaches that individually label proteins or lipids is not addressed by the authors, making it challenging to quantitatively address its performance, although it looks promising from the example data presented. The majority of the manuscript focuses on the utility of the approach to reveal new biology, which the authors center on contact sites and phase-separated organelles. This aspect of the study was less compelling and many of the observations are already well documented in the literature. As detailed in the points below, there is a lack of rigorous quantitative analysis, prior citation of observations of similar cell biological phenomena is inadequate, and the added value of land-ExM is not entirely clear.

Major Points:

1. More comparison to other ExM approaches for proteins or lipids, or using structures of known dimension, are needed to address the robustness and reproducibility of the expansion factor and the performance of land-ExM.
2. The criteria for what the authors deem a contact site, based on the land-ExM alone, are not sufficiently clear - the same is true for "phase separations". Quantitative metrics are needed to discriminate a contact site from two membranes that are in close proximity for other reasons including validation that would alter these contacts (for example genetically) in some way to validate the approach.
3. Tunnels through the nucleus/deep invaginations of the nuclear envelope have been described in many prior studies including from light and electron microscopy and these studies should be cited - indeed there are whole reviews on this topic including PMIDs 21514163 and 39367888). I also would suggest that the authors use the term tunnel if they pass all the way from one side to the other (usually between basal to apical surfaces of the nucleus) versus an invagination, which does not. Given the

whole volumes they can visualize this distinction should be possible.

4. If the authors wish to infer an important functional connection between the nuclear tunnel and nucleoli then it is imperative that they show a statistical over-representation of their proximity. Even if this is the case, the data as presented do not demonstrate a "membrane-phase separation contact site" - this is a possibility but based on the evidence presented it is an over-interpretation. Indeed, it could be that the tunnel needs to deform around the nucleolus, giving the impression of a specific interaction. More generally, the authors suggest that land-ExM is generally an approach that can reveal the relationship between phase condensates and membranes. While this is potentially the case, orthogonal approaches would certainly be required to make a strong argument for such interfaces. Another obvious question is whether these structures are observed in other cell types (also the observations presented in Fig. 3) - given the literature on such channels this is likely (e.g. A549 lung adenocarcinoma cells) - but it is unclear (at least to my knowledge) whether such structures are observed in non-transformed cells.

5. What element(s) of the cytoskeleton can the authors discern using land-ExM? The general term "cytoskeleton" in Fig. 2G (Line 176). Given the authors' statement that their approach is compatible with IF, the performance with regards to observing the cytoskeleton, a limitation of other approaches, would be of importance to many potential users of the method.

6. While the images in Figure 3 are very provocative and possibly were only observed because of the use of land-ExM, it is imperative to understand the frequency of the relationship between G3BP1 and these nuclear channels in order to be confident in the robustness of the observation given that the Figure only features a single example.

7. A related criticism to Point 6 is in regards to the relationship between the nuclear tunnels. Although the accumulation of G3BP1 is indeed exciting, given G3BP1 participates in other functions it would be necessary to define whether these are in fact stress granules. More over, single stress granules should be readily observed in the mid-plane of the nucleus should this be the case, additional evidence to support this point would strengthen the authors' claims. Last, it is not clear whether this observation in fact represents anything beyond the previous relationship described between the ER (which is largely functionally equivalent to the outer nuclear membrane) and SGs as described in the prior studies as cited by the authors.

8. This point of the authors was puzzling: "Intriguingly, we also observed the endoplasmic reticulum (ER) located between stress granules (SGs), nuclear invaginations, and nucleoli. In our four-color land-ExM images, which captured lipids, proteins, and immunostained SG and ER markers (Fig. 3L), the ER was found to overlap with the membrane of nuclear tunnels and adjacent to SGs (Fig. 3M-P)" (Lines 226-229). The outer nuclear membrane, which would necessarily line such nuclear channels, has largely the same composition as ER and will therefore contain Sec61. Therefore the "nuclear invaginations" and "ER" are expected to be one and the same unless the authors can show the presence of additional membrane bilayers, which do not appear to be visualizable in the lipid staining in Fig. 3 M-Q. Therefore I do not think that the discussion in Lines 230-238 is appropriate. An estimate of the prevalence beyond this one example is also not reported.

9. The authors state "Fig. 4 demonstrates how land-ExM revealed the MCSs between mitochondria and nuclear invaginations, a phenomenon that has been rarely reported" (Lines 262-263). First, there are many reports of contact sites between the outer nuclear membrane/nuclear envelope and mitochondria, and these should be cited (as just one recent example published in JCB see PMID 34694322). Second, the observation is that mitochondria are close apposed to the nuclear channels (at least in this one example - again there is no estimate of the prevalence in a population of cells), but it unclear both what the meaning of this observation may be nor whether this is an observation that could not be made conventional light microscopy.

Minor Points:

1. I would not consider nuclear invaginations/tunnels to be organelles - I think that the authors need to use a different term if they want to include these structures (Line 86).

2. Line 50-51. Clarification is needed for this sentence as the meaning of the "fold" value is not clear - this should provide a clear "effective" resolution. "ExM enables light microscopes to achieve 3- to 20-fold effective resolution" - presumably this is compared to the diffraction limit in conventional light microscopy?

3. While G3BP1 is indeed well known for its association with stress granules, it also has roles at membraneous organelles, for example damaged lysosomes. Given the arsenite treatment to induce stress granules, it is possible that this localization is related to these prior studies, which should be cited and discussed.

Authors' Responses to Comments

JCB manuscript #: 202502035

Title: Landscape Expansion Microscopy Reveals Interactions between Membrane and Phase-Separated Organelles

Authors: Yinyin Zhuang, Zhao Zhang, Zhipeng Dai, Xiaoyu Shi

Date: July 26, 2025

We thank the editors and all reviewers for their in-depth reading of our manuscript and for the insightful and constructive comments and suggestions. We have been intellectually stimulated by these comments while doing experiments, composing the authors' responses, and revising the manuscript. In this letter, we address the general comments and each point in detail. Through this letter, the authors' responses are in black and the quoted editor and all reviewers' comments are in blue. All changes in the revised manuscript and SI are highlighted in yellow.

Editors' comments:

You will see that while the Reviewers feel that land-ExM could be a promising method they also raise substantial concerns about lack of sufficient comparison with existing expansion microscopy methods and a clear demonstration that land-ExM provides an advantage and reveals structures that existing methods do not. The Reviewers also pointed out an absence of quantitative analysis of the observed contact sites and condensates as well as limited novelty provided by the imaging of nuclear invaginations.

However, given interest in the topic, we would be open to resubmission to JCB of a significantly revised and extended manuscript. A revised paper would most likely be best suited for the JCB Tools format. This would require refocusing the work to make it clear that the key advance is the adaptation of existing anchors to improve simultaneous imaging of proteins and lipids and provide comparisons with existing methods. Additionally, a more quantitative analysis of the nuclear invaginations and their potential contacts or proximity with other organelles and condensates as well as showing that land-ExM is a better option for imaging of these types of structures would be necessary.

Response to editors' comments:

We appreciate the opportunity to submit a substantially revised manuscript in the JCB Tools format. As the editors advised, we refocused the work to make it clear that the key advance is the adaptation of existing anchors to improve the simultaneous imaging of proteins and lipids, and to provide comparisons with existing methods. The revised manuscript focuses tightly on new techniques for membrane-phase separation contact, rather than on biological discovery. The key points of the revised manuscript are: (1) that the adaptation of our existing trifunctional anchors in expansion microscopy improves simultaneous imaging of proteins and lipids; (2) that land-ExM is optimized for visualization of contact/proximity of membrane and phase-separated protein organelles.

Here, we summarize the major revisions we've done to meet the requirements for resubmission the manuscripts in the in the JCB Tools format. First, we applied the land-ExM to pan-ExM and TReX and compared the results with the original pan-ExM and TReX, to address the concerns about lack of sufficient comparison with existing expansion microscopy methods. Second, we used these results to clearly demonstrate that land-ExM provides an advantage on revealing structures that existing methods do not. Third, we conducted more quantitative analysis on the nuclear tunnels and their potential contacts or proximity with condensates and ER. Collectively, we show that land-ExM is a better option for imaging contact sites between membrane organelles and phase separation.

Responses to reviewers' comments:

Reviewer #1:

This manuscript introduces Landscape Expansion Microscopy (Land-ExM), which incorporates a trifunctional anchoring strategy (NHS-biotin-MA) to improve biomolecule retention in expansion microscopy (ExM), particularly for proteins and lipids. The authors claim that this anchoring strategy enhances fluorescence signal intensity and enables high-resolution visualization of membrane contact sites (MCSs) and phase-separated organelles, achieving an effective resolution of ~30 nm with Airyscan microscopy. They demonstrate Land-ExM through imaging quadruple-organelle contact sites and mitochondrial-nuclear interactions, which they suggest are challenging to observe using conventional ExM techniques.

While the proposed anchoring strategy improves fluorescence intensity, the manuscript does not convincingly establish Land-ExM as a significant advancement over existing ExM methods. The resolution claim is misleading because the 30 nm resolution is due to the combination of 4× Expansion and Airyscan microscopy rather than the expansion process itself, meaning the intrinsic resolution remains ~60-70 nm using diffraction-limited microscopes, similar to conventional 4× ExM. There is no direct benchmarking against other high-resolution ExM techniques, such as TReX, iExM, Pan-ExM, or Expansion Revealing, which already achieve comparable or superior results through different expansion factors or anchoring strategies. Furthermore, the demonstration experiments fail to establish that the observed organelle interactions are uniquely resolvable by Land-ExM, as the previously mentioned ExM methods combined with mCLING pre-staining could likely yield the same or better results. Additionally, NHS-biotin-MA should theoretically have similar immunofluorescence signal retention as MA-NHS anchoring method, which has been reported by Chozinski et al. in 2016, making it unclear whether Land-ExM provides any real advantage for antibody-based labeling. Given these issues, the manuscript requires major revisions to properly validate its claims through comparative studies, clarify its resolution improvements, and provide evidence that it offers distinct advantages over existing techniques. However, even with these improvements, the contribution may be more appropriate for a specialized microscopy or bioimaging journal rather than a broad high-impact venue.

General response:

The reviewer's general comments focus on two concerns. The first concern is that land-ExM does not have an advantage in improving the resolution that previous expansion microscopy methods provide. We agree with the reviewer and would like to clarify that the land-ExM was not designed for a higher resolution. The resolution of existing expansion microscopy methods is outstanding for most biological applications. The goal of developing land-ExM is to achieve a higher signal-to-noise ratio (SNR) of existing expansion microscopy methods. Specifically, we wanted high SNR in both lipid and protein channels to allow discoveries of contact between membrane and phase-separated protein organelles, such as nuclear invagination and stress granule, and ER and stress granules.

The second concern is the lack of comparison with existing methods to show that the anchoring strategy of land-ExM improves the signal. We conducted experiments to compare land-ExM with TReX and pan-ExM following the reviewer's advice, which validated that the advantage of land-ExM in elevating the signal in lipid and protein imaging. We conducted a side-by-side comparison between land-ExM and pan-ExM, and between land-ExM and TReX. The new Figure 2 summarizes the results of the comparison, showing that land-ExM achieves higher SNR in both lipid and protein channels than those of pan-ExM and TReX.

The third concern is that "NHS-biotin-MA should theoretically have similar immunofluorescence signal retention as MA-NHS anchoring method, which has been reported by Chozinski et al. in 2016, making it unclear whether Land-ExM provides any real advantage for antibody-based labeling". The anchoring

efficiency of NHS-biotin-MA and MA-NHS should be equivalent. However, the biotin as the reporter theoretically retains more signal than additional dye-NHS. If the dye-NHS is added before anchoring and gelation, more 40% of the dye will lose fluorescence during polymerization and denaturation (Tillberg et al, 2016). NHS-biotin-MA uses biotin as the reporter, which remains unaffected by polymerization and denaturation, and is fluorescently labeled by STV-dye for imaging. If the dye-NHS is added after anchoring and gelation, the protein primary amines available for dye-NHS to react with are largely reduced by the MA-NHS anchoring. On the contrary, NHS-biotin-MA has the reporter and anchor in the same molecule and does not compete for the primary amines of proteins. In either workflow, the trifunctional anchors, NHS-biotin-MA, will provide higher signals. Our experimental results proved this theory. We will unfold the results in the point-by-point responses as follows.

Point-to-point responses:

1. The resolution claim of ~30 nm is not due to Land-ExM's expansion process itself but rather largely to the Airyscan microscopy, which technically is a super-resolution microscope at its own right. Since the expansion factor remains at 4×, the actual improvement in spatial resolution is limited to ~60-70 nm under diffraction-limit microscopes, similar to other 4× ExM methods. The authors should clarify that Land-ExM does not inherently improve spatial resolution beyond conventional 4× ExM and avoid misleading comparisons with high-expansion techniques like 10× ExM or iExM.

We thank the reviewer for pointing out the confusing message in our original manuscript. In the revised manuscript, we clarified that the 30 nm resolution is the outcome of a combination of 4x expansion and Airyscan, not anything specific about land-ExM. The goal of developing land-ExM is not to improve the resolution of existing expansion methods, but to improve the signals for contextual expansion microscopy methods. In our original abstract, we described this idea as “*Although expansion microscopy (ExM) provides superresolution, obtaining high signal-to-noise images for both proteins and lipids remains challenging. Land-ExM overcomes this limitation by using self-retention trifunctional anchors to significantly enhance protein and lipid signals in expanded samples.*” In the revision, 4th paragraph of the Results section, we further clarified this idea in the main text when we first time discussed the resolution of land-ExM:

“The measured lateral resolution of the Airyscan microscope is 138 nm. The 4.0 linear expansion factor of the cells used for Fig. 1 results in an effective lateral resolution of 35 nm. With this resolution and land-ExM’s high signal-to-noise ratio, contact sites between the nuclear tunnel and the nucleolus were clearly visualized (Fig. 1K&L). This observation is consistent with previous findings using electron microscopy (Bouteille and Hemon, 1979; Malhas et al., 2011). Compared with electron microscopy, land-ExM’s faster speed, 3D imaging, and multiplexity will enable a more statistical understanding of the interactions between the nuclear tunnel and the nucleolus. We explored the frequency and functions of the nuclear tunnel-nucleolus interaction with land-ExM in a recent preprint (Zhuang et al., 2024).”

2. The manuscript lacks a rigorous comparison with other ExM methods, making it difficult to assess whether Land-ExM provides a meaningful advantage. The only quantitative comparison focuses on fluorescence intensity using SYPRO Orange and NHS-Alexa Fluor 488, but these reagents were not used in a way that aligns with standard ExM workflows. Typically, SYPRO Orange and NHS-Alexa Fluor 488 are applied post-expansion, not before expansion, to avoid fluorophore degradation during the gelling process. Additionally, SYPRO Orange does not covalently bind to proteins, meaning it is largely washed away during homogenization, which explains its weak signal in the figure. However, this weak signal is misleading, as it does not accurately reflect the retention capabilities of conventional ExM approaches. Neither SYPRO Orange nor NHS-Alexa Fluor 488 are anchoring reagents and thus comparing them directly with Land-ExM's trifunctional anchor does not provide meaningful insight into biomolecule retention. A proper comparison would use MA-NHS as the anchoring reagent for the control group (PMID: 27064647) and then apply SYPRO Orange and NHS-Alexa Fluor 488 post-expansion, which would allow for an

accurate assessment of Land-ExM's effectiveness. Without this, it remains unclear whether Land-ExM truly improves biomolecule retention beyond what existing methods offer.

We thank the reviewer for their thoughtful strategies on the fair comparison. We conducted new experiments to compare land-ExM with pan-ExM and TREx, two commonly used contextual ExM methods. We strictly followed the protocols of pan-ExM and TREx. Our results showed that the land-ExM provides significantly higher signals in both protein and lipid channels, compared with pan-ExM and TREx. These results construct the new Figures 2B-M in our revised manuscript, also showing below:

Fig. 2. Land-ExM labeling and anchoring strategies improve the signal of TREx and pan-ExM. (A) Workflow of land-pan-ExM, which only replaces the labeling strategy of pan-ExM with the labeling strategy of land-ExM. (B) land-TREx protein channel of U2OS cells, where proteins were labeled and anchored with NHS-Biotin-MA. Scale bar: 5 μ m in pre-expansion unit. Linear expansion factor: 7. (C) TREx protein channel of U2OS cells, where proteins were anchored with acryloyl-X SE and stained with Alexa Fluor 488 NHS ester. Scale bar: 5 μ m in pre-expansion unit. Linear expansion factor: 7. (D) Bar chart comparing the signal-to-noise ratio of the protein channel in land-TREx and TREx. The signal-to-noise ratio is calculated as the average pixel value of the area with cells divided by the average pixel value of the area without cells in each image. Each bar represents the mean \pm standard error of more than 20 cells. (E) land-TREx lipid channel of U2OS cells, where lipids were labeled by mCLING and anchored with NHS-Biotin-MA. Scale bar: 5 μ m in pre-expansion unit. Linear expansion factor: 7.0. (F) TREx lipid channel of U2OS cells, where lipids were anchored with acryloyl-X SE and stained with mCLING. Scale bar: 5 μ m in pre-expansion unit. Linear expansion factor: 7.0. (G) Bar chart comparing the signal-to-noise ratio of the lipid channel of land-TREx and TREx. The signal-to-noise ratio is calculated as the average pixel value of the area with cells divided by the average pixel value of the area without cells in each image. Each bar represents the mean \pm standard error of more than 20 cells. (H) land-pan-ExM protein channel of U2OS cells, where proteins were labeled and anchored with NHS-Biotin-MA. Scale bar: 5 μ m in pre-expansion unit. Linear expansion factor: 12.0. (I) pan-ExM protein channel of U2OS cells labeled with Alexa Fluor 488 NHS ester. Scale bar: 5 μ m in pre-expansion unit. Linear expansion factor:

12.0. (J) Bar chart comparing the signal-to-noise ratio of the protein channel in land-pan-ExM and pan-ExM. The signal-to-noise ratio is calculated as the average pixel value of the area with cells divided by the average pixel value of the area without cells in each image. Each bar represents the mean \pm standard error of more than 20 cells. (K) land-pan-ExM lipid channel of U2OS cells, where lipids were stained following the workflow (A). Scale bar: 5 μ m in pre-expansion unit. Linear expansion factor: 12.0. (L) pan-ExM lipid channel of U2OS cells labeled with mCLING. Scale bar: 5 μ m in pre-expansion unit. Linear expansion factor: 12.0. (M) Bar chart comparing the signal-to-noise ratio of the lipid (mCLING) channel in land-pan-ExM and pan-ExM. The signal-to-noise ratio is calculated as the average pixel value of the area with cells divided by the average pixel value of the area without cells in each image. Each bar represents the mean \pm standard error of more than 20 cells. All images were taken with an Airyscan microscope.

Additionally, in our original manuscript, we did exactly what the reviewer suggested: “A proper comparison would use MA-NHS as the anchoring reagent for the control group (PMID: 27064647) and then apply SYPRO Orange and NHS-Alexa Fluor 488 post-expansion, which would allow for an accurate assessment of Land-ExM’s effectiveness.” In the original manuscript, we described it as “*The workflows for these control groups were identical to land-ExM except for steps (4) and (7), where methacrylic acid N-hydroxysuccinimide ester (NHS-MA) was used as the protein anchor in step (4), and proteins were stained with NHS-Alexa Fluor 488 or SYPRO Orange in step (7).*” We hope this clarification will make it clear to the reviewer that land-ExM truly improves biomolecule retention.

3. The manuscript does not provide data comparing immunofluorescence (IF) efficiency, making it unclear whether Land-ExM offers any real advantage in retaining antibody-based labels. In principle, this anchoring approach is not expected to improve IF retention compared to MA-NHS anchoring, as the biotin-streptavidin amplification strategy does not enhance antibody labeling, which already relies on covalent binding or strong antigen-antibody interactions. Additionally, MA-NHS anchoring method has been reported by Chozinski et al. in 2016, which further decreases the novelty of this work. Without direct comparisons of pre-expansion vs. post-expansion immunofluorescence signal intensity between Land-ExM and a standard NHS-MA anchoring protocol, there is no evidence that Land-ExM improves IF-based ExM workflows. Similarly, the study lacks a quantitative comparison of lipid probe retention efficiency, particularly for mCLING, which is a key marker for membrane structures. Since mCLING is already biotinylated, the use of streptavidin in Land-ExM could simply enhance its signal rather than genuinely improving lipid retention, making it unclear whether this anchoring approach provides any retention benefits over standard lipid-labeling strategies. To properly evaluate Land-ExM’s performance, the authors should provide quantitative retention efficiency comparisons for both immunofluorescence signals and lipid probes (e.g., mCLING) against conventional ExM anchoring methods.

In response to the reviewer’s question on if land-ExM offers any real advantages in retaining antibody-based labels compared with MA-NHS anchoring method, we refer to our paper on Label-Retention Expansion Microscopy (Shi et al, *JCB* (2021) 220 (9): e202105067). In this 2021 paper, we proved that the trifunctional anchors (e.g. MHS-biotin-MA and NHS-DIG-MA) offer a real advantage in retaining the signal of IF-based ExM, compared with the MA-NHS anchoring method (Chozinski et al. in 2016, and proExM by Tillberg et al., 2016). The figure below (part of Figure 1, *JCB* 2021) shows using antibodies conjugated with trifunctional anchors MHS-biotin-MA (E, quantified as the right column in F) can significantly increase fluorescence retention after expansion, compared with antibodies conjugated with organic dye and anchored with MA-NHS (C, quantified as the left column in F). Comparing the NHS-biotin-MA, the NHS-DIG-MA further amplifies the signal (G) because each anti-

DIG antibody has multiple dyes. Therefore, we chose antibodies conjugated with MA-biotin-DIG for IF to pair with land-ExM in this manuscript.

The reasons that antibody conjugated with MA-biotin-NHS retains higher signal than antibody conjugated with organic dye and anchored by MA-NHS are: (1) that about 50% of organic dye react with the radicals generated during the polymerization and go to dark states (Tillberg et al. 2016), and (2) that the organic dye on an antibody can be washed out due to accidental antibody fragmentation during the homogenization step. The trifunctional protein probe, MA-biotin-NHS, used in land-ExM is not fluorescent and thus not bleached by radicals during polymerization; it also self-anchors to the polyacrylamide chains and is not washed away by protein fragmentation. The reason (2) matters more for proteinase K digestion than for heat denaturation. This is why most of the protein/lipid contextual expansion microscopy methods use heat denaturation, such as pan-ExM, TReX, and umExM. However, since land-ExM covalently anchors protein labels, lipid labels, and IF labels to the hydrogel, their signals are resistant to proteinase K digestion. We did new experiments to prove the land-ExM's compatibility with ExM protocols using proteinase K. Here we show the results:

Figure S3. land-ExM using proteinase K digestion. (A) Workflow of land-ExM using proteinase K digestion to homogenize cells instead of heat denaturation. (B) land-ExM protein image of U2OS cells with proteinase K digestion (proK). Scale bar: 10 μm in pre-expansion unit. Linear expansion factor: 4.0. (C) land-ExM protein image of U2OS cells with heat denaturation (heat). Scale bar: 10 μm in pre-expansion unit. Linear expansion factor: 4.0. (D) land-ExM lipid image of U2OS cell with proteinase K digestion (proK). Scale bar: 10 μm in pre-expansion unit. Linear expansion factor: 4.0. (E) Land-ExM lipid image of U2OS cells

with heat denaturation (heat). Scale bar: 10 μm in pre-expansion unit. Linear expansion factor: 4.0. All images are taken with an Airyscan microscope.

Response to the reviewer's comment on lacking a quantitative comparison of mCLING retention efficiency: We agree with the reviewer that MA-NHS and NHS-biotin-MA should both anchor mCLING with similar efficiency. And we experimentally observed that MA-NHS and NHS-biotin-MA worked equally well for lipid mCLING imaging. We described this point in the 5th paragraph in the Results section of our original manuscript as follows. "For users focusing on lipid context but with a low requirement for protein imaging, commercial NHS-MA or glycidyl methacrylate (GMA) (Cui et al., 2023) can serve as an alternative to NHS-biotin-MA (Fig. 1A, step 4), and NHS dyes can be used for protein fluorescent labeling in step 7. This will yield an equally high signal-to-noise ratio in lipid imaging following our instructions in the Methods section, but with significantly dimmer signals in the protein channel (Fig. 1G)."

4. The claim that Land-ExM uniquely enables the visualization of mitochondrial-nuclear invagination contacts is not supported. Other ExM methods, particularly those with higher expansion factors (10 \times ExM, iExM, Pan-ExM, Expansion Revealing), could likely resolve these structures if the sample was pre-stained with mCLING. The authors do not provide evidence that these contacts could not have been observed with

alternative ExM approaches. To support their claim, they should either compare Land-ExM to another ExM method for MCS imaging or revise their statement to acknowledge that other methods could achieve similar results.

We thank the reviewer for pointing out the confusing statement. We didn't intend to claim that these mitochondrial-nuclear invagination contacts will only be discovered by land-ExM. We used this case to demonstrate the kinds of spatial relationships that land-ExM can be used to discover. To shift the focus of the paper to tool development, we deleted the section "Visualizing membrane contact sites" from the manuscript and added more technical sections, such as "*Land-ExM is compatible with proteinase K digestion*" and "*Land-ExM enhances protein and lipid signals of TReX and pan-ExM.*"

5. The streptavidin-biotin labeling strategy introduces potential artifacts, as fluorophore-bearing streptavidin binds to both biotinylated proteins and biotinylated lipid probes (e.g., mCLING), which could cause non-specific signal overlap.

The reviewer's sharp chemical picture caught a potential flaw. We had the same concern when developing land-ExM. But we experimentally proved that the crosstalk is negligible. We discussed it in the 5th paragraph in the Results section of our original manuscript as follows: "As NHS-biotin-MA reacts to both proteins and mCLING, there is a potential risk of cross-contamination between protein and lipid signals. However, this contamination is negligible due to the significantly lower abundance of mCLING than native proteins. This is validated by the distinct fluorescence patterns in protein and lipid channels (Fig. 2). The results were not shown in the original manuscript, but we included them in the revised.

During revision, we had a chance to develop a different approach to completely avoid the crosstalk between the protein and lipid channels, following Reviewer 2's suggestion: "first do the MA-biotin-NHS step and then the mCLING step (with another GMA or MA-NHS step at the end to retain mCLING if necessary) to avoid any potential crosstalk". We implemented this approach and presented the results in a new supplementary figure (Fig. S2) shown below. This approach is discussed as an alternative land-ExM workflow in the revised manuscript. We compared the outcome of lipid and protein imaging using this new approach (Fig. S2B) with that using our original workflow (Fig. S2C). They are very similar to each other in all cell types we tested. This confirmed our original statement is true, "the crosstalk is negligible". However, we expect this new workflow to work better in samples exceptionally rich in lipid.

Fig. S2. Alternative land-ExM workflow to avoid crosstalk between

NHS-Biotin-MA and mCLING. (A) Alternative workflow of land-ExM. (B) i and ii: land-ExM images of U2OS cells stained first with NHS-Biotin-MA then mCLING. iii to v: Magnified images of boxes in i and ii. vi: Normalized intensity profile along the orange line in v. Scale bar: 5 μm in pre-expansion unit. Linear expansion factor: 4.0 (i and ii). 0.5 μm in pre-expansion unit. Linear expansion factor: 4.0 (iii to v). (C) i and ii: land-ExM images of U2OS cells stained first with mCLING then NHS-Biotin-MA. iii to v: Magnified images of orange boxes in i and ii. vi: Normalized intensity profile along the orange line in v. Scale bar: 5 μm in pre-expansion unit. Linear expansion factor: 4 (i and ii). 0.5 μm in pre-expansion unit. Linear expansion factor: 4.0 (iii to v). All images are taken with an Airyscan microscope.

Reviewer #2:

Zhuang et al., in their manuscript "Landscape Expansion Microscopy Reveals Interactions between Membrane and Phase-Separated organelles" describe a new microscopy technique, Landscape Expansion Microscopy (land-ExM) and demonstrate its application to multi-organelle contacts. Overall it is strong work and I recommend publication after addressing the minor points below.

Land-ExM is based on earlier work by Prof. Shi on label-retention expansion microscopy (LR-ExM) that uses trifunctional small-molecule probes containing an NHS ester group to label proteins nonspecifically together with the mCLING probe to label lipids. These can be combined with immunostaining to enable a number of novel experiments. Similar experiments were reported by Damstra et al., eLife 2022 (cited ref. 19). In those experiments, Damstra 2022 performed similar measurements using NHS ester-functionalized dyes to label proteins and mCLING to label lipids, however the authors of that paper commented that the protein and lipid signals were very similar to one another. In contrast, Zhuang et al. now report and improve protocol that achieves distinct lipid and protein signals, each of which is highly information rich. It is a valuable contribution with high-quality results and it will be of interest to a broad audience.

I have many minor points that can be addressed with simple revisions to the text or figures. In one case, a small measurement may be helpful, but I do not see this as required.

General response:

We are heartened to read, "It is a valuable contribution with high-quality results, and it will be of interest to a broad audience." It encourages us to keep doing solid work. We addressed all the reviewers' comments point by point as follows.

Point-by-point responses:

- Many acronyms are indicated in Fig. 1a. These should ALL be spelled out for clarity to the readers in the figure or caption, including PFA, GA, IF, STV. In addition, on panel A) step 5 the spelling should be "gelate".

We thank the review's close reading. In the revised Figure 1, we spelled out all the acronyms and corrected the typo to "gelate".

- Fig 1H contains beautiful data. I suggest that the authors show the positions where the different views correspond using, for instance, some dashed lines. This helps the reader gain a better appreciation of the 3D structure within their data.

We thank the reviewer for the advice, which improve the data presentation. We added dashed lines in x-y views in the revised Figure 1 to show the positions where orthogonal views correspond, as shown below.

• I think it has become a bit fashionable to display pan-ExM or similar NHS labeling type images in electron microscopy (EM) style with black on white to make a point about how that staining is conducted with low molecular specificity similar to EM. This is a purely stylistic point, but I'd encourage authors to think hard about what format is generally easiest for the reader to observe for the type of data overall and to stick with that (I suspect it is white on black). For instance, multichannel data with blue lipid and black protein signal in Fig. 1 don't particularly go well together since the overlay doesn't change the hue...for this reason, typically fluorescence uses magenta/green for two-channel data because a distinctive white overlay is easier to perceive. The authors later switch to bright on dark in Figures 3, S2, and S3 which I personally prefer. Figure 3 does put the protein signal as a white on black as I suggest, but white doesn't go super well with red. Why not just stick with standard 2- and 3-channel color schemes like MG or RGB or CMY?

We appreciate that the reviewer paid attention to the visual presentation. It is a challenging task to select colors to present multicolor images. As the reviewer found, we struggled with the black or white backgrounds and switched back and forth in our figures. In response to this comment, we compared various formats and asked our colleagues and students for their feedback as readers. We compared some of the formats we tried in the figure below. First, for single-color presentations, we compared the black signal on a white background (A) and the white signal on a black background (B). As the reviewer suspected, white on black worked better for more than 50% of readers. These readers found it easier to see more features at a middle brightness on a black background (D) than on a white background. However, researchers who usually process EM data primarily found it the other way around.

For multicolor images, we agree with the reviewer that magenta/green (or red/cyan) composite (F and H) on will reveal the spatial relationships between two channels better than blue/grey composite (E and G). For Figure 3, we agree that red and white are not the best color combination, and red/cyan is a standard 2-channel choice. We used the protein in white to ensure consistency of the colors throughout all panels in Figure 3, to avoid confusion for readers. We left the cyan color to the lipid channel.

• While I'm on the subject of figure format, I'll just mention that many people consider it a best practice when displaying separate channels (from a multicolor image) as grayscale images not color images. This avoids issues related to different levels of color perception. For instance see Jayme Johnson, MBoC, 2012 or the MBoC color guide <https://www.ascb.org/diversity-equity-and-inclusion/how-to-make-scientific-figures-accessible-to-readers-with-color-blindness/>. This is relevant for Figures S2 & S3.

We converted the separate channel images in Figures S2 & S3 to greyscale. And the two Figures are updated as Figures S4 and S5 in the revised manuscript.

Figure S4

Figure S5

- Figure 2 panel N would be improved by including a legend for the color bar to know what the heights are for the color bar (e.g., blue = 0 μ m and white = 6 μ m or something).

We added the “0” and “6 μ m” next to the color bar on the Figure 2 as shown below, and described it in the figure caption as “Color bar: purple to white: 0 to 6 μ m in pre-expansion unit.” **We were surprised that the reviewer precisely estimated the exact z range that we took on our microscope?!**

- The caption and text of Fig 2 is not clear regarding what signal is in what channel. The caption indicates that Land-ExM reveals things but as I understand it Land-ExM is a dual protein and lipid stain. I get the impression that Fig 2a-g shows the protein signal because it is black and white like what we saw in Fig 1, and then maybe H-M are lipid signal but this would be important to state in the figure and/or caption.

We thank the reviewer for catching this unclear description. Fig. 2A-G shows the protein signal, and H-M shows the lipid signal. We clarified the protein or lipid staining in the caption and text of Fig 2 (Fig 3 in the revised manuscript) accordingly. Clarification is marked in yellow in the section “Land-ExM identifies membrane organelles and phase separation based on morphology” and in the figure caption, copied below.

Fig. 3. Land-ExM identifies phase-separated and membrane organelles. (A-G) land-ExM protein images of membraneless phase separation structures. The proteins were labeled with NHS-biotin-MS, and post-gelation stained with streptavidin-Alexa Fluor 488. (A) land-ExM protein image of nucleoli in a U2OS cell. Red arrowheads indicate the fibrillar center (FC) or dense fibrillar component (DFC) of the nucleolus. Scale bar: 1 μ m in pre-expansion unit. Linear expansion factor: 4.0. (B) land-ExM protein image of nuclear bodies of breast cancer cell, UCI082014. Red arrowheads indicate the nuclear bodies. Scale bar: 1 μ m in pre-expansion unit. Linear expansion factor: 4.2. (C) land-ExM protein image of stress granules of a U2OS cell treated with NaAsO₂ for 20 minutes. The red arrowhead indicates a stress granule. Scale bar: 1 μ m in pre-expansion unit. Linear expansion factor: 4.0. (D) land-ExM protein image of chromatin of a breast cancer cell. Scale bar: 500 nm in pre-expansion unit. Linear expansion factor: 4.2. (E) land-ExM protein image of nuclear pore complexes of a breast cancer cell. Scale bar: 1 μ m in pre-expansion unit. Linear expansion factor: 4.2. (F and G) land-ExM protein images of mitochondria and cytoskeleton of a U2OS cell. Scale bar: 1 μ m in pre-expansion unit. Linear expansion factor: 4.0. (H-P) land-ExM lipid images of membrane structures. The lipids were labeled with mCLING-Atto647N. (H) land-ExM lipid image of breast cancer cell. Scale bar: 5 μ m in pre-expansion unit. Linear expansion factor: 4.0. (I-M) magnified images of (H) showing different membrane structures: lipid vesicles (I), mitochondria (J), filopodia (K), nuclear invagination (L), and Golgi apparatus (M). Scale bar: 1 μ m (I-M) in pre-expansion unit. (N) 3D land-ExM lipid image of a breast cancer cell after maximum intensity projection, showing the cell membrane. Color-coded by the z-dimension slices from bottom to top. Color bar: purple to white: 0 to 6 μ m in pre-expansion unit. Scale bar: 5 μ m in pre-expansion unit. Linear expansion factor: 4.0. (O and

P) magnified images of (N) showing detailed structures of the cell membrane. Scale bar: 1 μm in pre-expansion unit. All images were taken with an Airyscan microscope.

• Line 57 of the supplement should probably be for (P) not (H) in Fig S2. Same issue for the Fig. S3 caption. We have corrected these typos. And the figures have now been updated as Figure S4 and S5.

• The authors compare protein signal for their method and for cells treated first with MA-NHS and then At488-NHS, showing that their method produces $\sim 10\times$ more signal. One might expect a two-step procedure to be more sensible, with 1mM dye-NHS stain first, followed by the 25 mM MA-NHS step. We agree with the reviewer. When all proteins are well anchored to the hydrogel, under heat denaturation, the dye-NHS should be well retained at a similar level to the NHS-biotin-MA. The disadvantage of using dye-NHS is that a percentage of the dye would react with the radicals during the free radical chain reactions in the polymerization steps and stay dark.

• Can the authors first do the MA-biotin-NHS step and then the mCLING step (with another GMA or MA-NHS step at the end to retain mCLING if necessary) to avoid any potential crosstalk? The authors could address this with a comment around line 162 where the possibility of cross-contamination is described, although it may be helpful to perform a test of this (or to show one in the supplementary information, if they have already performed such a test).

We appreciated the reviewer's strategy that can completely avoid the crosstalk. We implemented this approach and presented the results in a new supplementary figure (Fig. S2) shown below. This approach is discussed as an alternative land-ExM workflow in the revised manuscript. We compared the outcome of lipid and protein imaging using this new approach (Fig. S2B) with that using our original workflow (Fig. S2C). They are very similar to each other in all cell types we tested. This confirmed our original statement is true, "the crosstalk is negligible".

However, we expect this new workflow to work better in samples that are exceptionally rich in lipid.

Fig. S2. Alternative land-ExM workflow to avoid crosstalk between NHS-Biotin-MA and mCLING. (A) Alternative workflow of land-ExM. (B) i and ii: land-ExM images of U2OS cells stained first with NHS-Biotin-MA then mCLING. iii to v: Magnified images of boxes in i and ii. vi: Normalized intensity profile along the orange line in v. Scale bar: 5 μm in pre-expansion unit. Linear expansion factor: 4.0 (i and ii). 0.5 μm in pre-expansion unit. Linear expansion factor: 4.0 (iii to v). (C) i and ii: land-ExM images of U2OS cells stained first with mCLING then NHS-Biotin-MA. iii to v: Magnified images of orange boxes in i and ii. vi: Normalized intensity profile along the orange line in v. Scale bar: 5 μm in pre-expansion unit. Linear expansion

factor: 4 (i and ii). 0.5 μm in pre-expansion unit. Linear expansion factor: 4.0 (iii to v). All images were taken with an Airyscan microscope.

- A spatial resolution of $\sim 30\text{nm}$ is asserted. While some line profiles are provided in Figures S2 & S3 the 30nm spatial resolution is not evident or commented on. This is not a major point. I suggest that the authors either show some evidence for 30 nm resolution or else simply suggest that a spatial resolution of 30 nm could be achieved with the method.

We thank the reviewer for raising this discussion. We characterized the resolution of the microscope with fluorescent beads, which was 138 nm. An image used to measure the resolution is shown below. In the revised main text, the 4th paragraph in the Results section, we described the effective resolution:

“The measured lateral resolution of the Airyscan microscope is 138 nm. The 4.0 linear expansion factor of the cells used for Fig. 1 results in an effective lateral resolution of 35 nm. With this resolution and Land-ExM’s high signal-to-noise ratio, contact sites between the nuclear tunnel and the nucleolus were clearly visualized (Fig. 1K&L).”

- On line 128 the authors estimate stoichiometry of dyes per NHS-biotin-MA molecule under the assumption that one streptavidin binds one NHS-biotin-MA group, however, streptavidin could bind multiple biotins if they were in sufficient proximity, albeit with different affinity.

We thank the reviewer for pointing out the physical picture of the streptavidin binding. We agree that “streptavidin could bind multiple biotins if they were in sufficient proximity, albeit with different affinity”.

- On line 132 the authors state that their method has significantly higher signal-to-noise ratio compared to ExM with Alexa Fluor 488. The authors showed significantly higher SIGNAL for their method (which is great) but they do not show any measurements of signal/noise. This could be fixed with a small text revision.

We thank the reviewer for pointing out the mismatch between what we aimed for (i.e., high SNR) and what we calculated (i.e., signal). The protein and lipid signals have been corrected to signal-to-noise ratio in the new Figure 1G, Figure 2D, J, G, and M. The signal-to-noise ratio is calculated as the average pixel value of the area with cells divided by the average pixel value of the area without cells in each image.

- I’m confused around line 265 why mitochondria are re-visited and sort of validated when that seemed to have already been done around line 178. Does it make sense to do this twice?

We agree it is redundant. We deleted the description around line 265. To shift the focus of the paper to tool development, we deleted the section “Visualizing membrane contact sites” from the manuscript, while adding more technical sections, such as “Land-ExM is compatible with proteinase K digestion” and “Land-ExM enhances protein and lipid signals of TReX and pan-ExM”.

- Around line 330, the authors might clarify slightly that they mean "Utilizing land-ExM labeling together with super-resolution techniques...", if this is indeed the case.

We revised the statement in the revised manuscript as follows:

"It is technically possible to push the resolution of land-ExM slightly beyond 10 nm by imaging with single-molecule localization microscopes, like STORM (Shi et al., 2021) and MINFLUX (Balzarotti et al., 2017; Schmidt et al., 2021), or employing more swellable hydrogel (Chang et al., 2017b),(Wang et al., 2024),(Shaib et al., 2024). However, the ultimate resolution that ExM can achieve is limited by the pore size of the hydrogel, and the nanoscale distortion caused by expansion must be carefully examined."

- Line 53 of supplement should say "Linear expansion."

We corrected the typo "liner expansion" to "linear expansion". And changed all "length expansion factors" to "linear expansion factors" to be consistent.

- The authors may wish to also cite the 2019 lipid expansion microscopy bioRxiv manuscript from the Boyden group at the same place where they cite the 2024 bioRxiv (cited reference 33).

We cited this preprint in the revised manuscript.

- The authors show in Fig. 3 a fascinating observation with nuclear invaginations and multiple organelle contacts. I see this paper as primarily a method paper that also demonstrates utility on an application, but that the number of cells and such that are analyzed for the various organelle contacts is small (1-2 cells?) so I wonder whether it would be more appropriate to de-emphasize the biological conclusions and to make it clear that the new tool facilitates observations. -- For instance, on line 220, the authors indicate "This spatial relationship was commonly seen in nuclear invaginations of cells under stress in our experiments." Similarly, on line 229 the authors indicate "ER-SG contacts inside nuclear tunnels were frequently observed in cells under stress in our experiments." But in each case, only one cell was shown or apparently scored for these phenotypes. -- I would suggest the authors determine whether they want to change the tone and scope to say that these features can be easily observed using this new method, or if they want to present some statistics or quantification on the phenotype such as how often it is observed in the treated vs control cells.

We agree with the reviewer regarding the focus of this manuscript. We refocused the manuscript on the method itself, as described up front in the Responses to Editors' Comments. We added new sections about various workflows of land-ExM, rephrased and adjusted the tone to de-emphasize the biological conclusions. Additionally, we have included more statistical analysis in the new Fig. 4R&S and Figure S7 (shown below), along with related text, to demonstrate that this spatial relationship is common in cells under stress.

Fig. 4R&S (R) Pie chart of nuclear tunnels with or without SGs. Total tunnels analyzed: 114. (S) Pie chart of SG-filled nuclear tunnels that contact nucleoli versus those that do not. Total tunnel analyzed: 83.

Figure S7. Land-ExM reveals stress granules at different locations of cells.

(A-C) Land-ExM images of U2OS cells untreated or treated with NaAsO₂ for 20 min or 60 min, then immunostained with anti-G3BP1 antibody. Scale bar: 5 µm in pre-expansion unit. Linear expansion factor: 4.

(D-F) Land-ExM images of U2OS cells stained with mCLING (magenta), NHS ester dye (cyan), and immunostained with anti-G3BP1 (yellow) and anti-Sec61b (white) antibodies. Cells were untreated or treated with NaAsO₂ for 20 min or 60 min. Scale bar: 5 µm in pre-expansion unit. Linear expansion factor: 4.

(G) Magnified images of (E) showing stress granules (SG) formed adjacent to ER (orange arrowheads). Scale bar: 1 µm in pre-expansion unit.

(H) Analysis of the number of nuclear tunnels per cell with or without 60 min

NaAsO₂ treatment. Each bar represents the mean ± standard error of more than 18 cells. The ns indicates $p > 0.05$ by Welch's t test.

(I) Analysis of the diameter of nuclear tunnels in cells with or without 60 min NaAsO₂ treatment. Each bar represents the mean ± standard error of more than 20 cells. ns indicates $p > 0.05$ by Welch's t test.

All images were taken with an Airyscan microscope.

Reviewer #3:

Summary: In this manuscript the authors describe an approach that they term landscape expansion microscopy or "land-ExM", which uses trifunctional labels that self-anchor to the gel that allows for 4-fold expansion upon hydration, which they combine with super-resolution microscopy. Land-ExM is an extension of their previously published approach termed label-retention (LR) ExM - the major new aspect of this new work is the addition of lipid labeling using the mCLING reagent. Applying the approach to U2OS cells, the authors report several observations including tunnels through the nucleus that are frequently near nucleoli, that contain labeling for Sec61, and that have stress granule components within them in response to stress and are often found in proximity to mitochondria at the nuclear surface.

Overall assessment: Technically the land-ExM approach appears to be capable of combining total protein and lipid information and is compatible with immunofluorescence microscopy. Membrane labeling has remained a major challenge in expansion microscopy until the past few years and represents an important aspect of future developments. How well land-ExM performs compared to other approaches that individually label proteins or lipids is not addressed by the authors, making it challenging to quantitatively address its performance, although it looks promising from the example data presented. The majority of the manuscript focuses on the utility of the approach to reveal new biology, which the authors center on contact

sites and phase-separated organelles. This aspect of the study was less compelling and many of the observations are already well documented in the literature. As detailed in the points below, there is a lack of rigorous quantitative analysis, prior citation of observations of similar cell biological phenomena is inadequate, and the added value of land-ExM is not entirely clear.

General response:

We thank the reviewer for their advice on acquiring more quantitative comparisons between land-ExM and other ExM methods. In response, we conducted new experiments to quantitatively compare the signal-to-noise ratio of land-ExM with that of TREx and pan-ExM. The results showed that land-ExM significantly increases the signals compared with TREx and pan-ExM. The results are shown in the new Figure 2.

During revision, we also addressed the reviewer's requests for clearer references and expanded quantitative analysis. We admit that the scope of this work is insufficient to include enough controls and gene editing for a comprehensive understanding of the biological consequences of the interplay of stress granules, nuclear invaginations, mitochondria, and ER. Therefore, we focused the revised manuscript on method development and de-emphasized the biological conclusions.

Point-by-point responses:

Major Points:

1. More comparison to other ExM approaches for proteins or lipids, or using structures of known dimension, are needed to address the robustness and reproducibility of the expansion factor and the performance of land-ExM.

We thank the reviewer for their advice, which is also shared in the comments of other reviewers. During revision, we conducted new experiments to compare the signal of land-ExM with that of pan-ExM and TREx. We strictly followed the protocols of pan-ExM and TREx and compared them with land-ExM. Our results showed that the land-ExM provides significantly higher signals in both protein and lipid channels, compared with pan-ExM and TREx. These results construct the new Figures 2B-M in our revised manuscript, also showing below:

Fig. 2. Land-ExM labeling and anchoring strategies improve the signal of TREx and pan-ExM. (A) Workflow of land-pan-ExM, which only replaces the labeling strategy of pan-ExM with the labeling strategy of land-ExM. (B) land-TREx protein channel of U2OS cells, where proteins were labeled and anchored with NHS-Biotin-MA. Scale bar: 5 μ m in pre-expansion unit. Linear expansion factor: 7. (C) TREx protein channel of U2OS cells, where proteins were anchored with acryloyl-X SE and stained with Alexa Fluor 488 NHS ester. Scale bar: 5 μ m in pre-expansion unit. Linear expansion factor: 7. (D) Bar chart comparing the signal-to-noise ratio of the protein channel in land-TREx and TREx. The signal-to-noise ratio is calculated as the

average pixel value of the area with cells divided by the average pixel value of the area without cells in each image. Each bar represents the mean \pm standard error of more than 20 cells. (E) land-TREx lipid channel of U2OS cells, where lipids were labeled by mCLING and anchored with NHS-Biotin-MA. Scale bar: 5 μ m in pre-expansion unit. Linear expansion factor: 7.0. (F) TREx lipid channel of U2OS cells, where lipids were anchored with acryloyl-X SE and stained with mCLING. Scale bar: 5 μ m in pre-expansion unit. Linear expansion factor: 7.0. (G) Bar chart comparing the signal-to-noise ratio of the lipid channel of land-TREx and TREx. The signal-to-noise ratio is calculated as the average pixel value of the area with cells divided by the average pixel value of the area without cells in each image. Each bar represents the mean \pm standard error of more than 20 cells. (H) land-pan-ExM protein channel of U2OS cells, where proteins were labeled and anchored with NHS-Biotin-MA. Scale bar: 5 μ m in pre-expansion unit. Linear expansion factor: 12.0. (I) pan-ExM protein channel of U2OS cells labeled with Alexa Fluor 488 NHS ester. Scale bar: 5 μ m in pre-expansion unit. Linear expansion factor: 12.0. (J) Bar chart comparing the signal-to-noise ratio of the protein channel in land-pan-ExM and pan-ExM. The signal-to-noise ratio is calculated as the average pixel value of the area with cells divided by the average pixel value of the area without cells in each image. Each bar represents the mean \pm standard error of more than 20 cells. (K) land-pan-ExM lipid channel of U2OS cells, where lipids were stained following the workflow (A). Scale bar: 5 μ m in pre-expansion unit. Linear expansion factor: 12.0. (L) pan-ExM lipid channel of U2OS cells labeled with mCLING. Scale bar: 5 μ m in pre-expansion unit. Linear expansion factor: 12.0. (M) Bar chart comparing the signal-to-noise ratio of the lipid (mCLING) channel in land-pan-ExM and pan-ExM. The signal-to-noise ratio is calculated as the average pixel value of the area with cells divided by the average pixel value of the area without cells in each image. Each bar represents the mean \pm standard error of more than 20 cells. All images were taken with an Airyscan microscope.

2. The criteria for what the authors deem a contact site, based on the land-ExM alone, are not sufficiently clear - the same is true for "phase separations". Quantitative metrics are needed to discriminate a contact site from two membranes that are in close proximity for other reasons including validation that would alter these contacts (for example genetically) in some way to validate the approach.

We thank the reviewer for pointing out the vague definition of "contact site" and "phase separation" in the original manuscript. We clarified them with quantitative metrics in the revised manuscript. Previous

literature defined a membrane contact site (MCS) as a specialized region where the membranes of two organelles come into close proximity, typically separated by only 10-80 nanometers (Scorrano et al, Coming together to define membrane contact sites. Nat Commun 10, 1287, 2019; Raimondi et al, Methods in Cell Biology, Chapter 5, 177, 2023). The distance between the membranes of two organelles has been used as the metric to discriminate a contact site from two membranes. Following the definition of MCS, we defined a contact site as two organelles (regardless of membrane or membraneless) come into close proximity, separated by less than 80 nanometers. And we used the distance between the membrane and the edge of a membraneless protein condensates (phase separation) as the metric to discriminate a contact site. The distance is measured by land-ExM. The land-ExM provides a resolution up to 12 nm. This resolution is sufficient to measure distance from 24 nm to 80 nm. Any contact sites with a distance smaller than 80 nm will appear overlapped/unseparable in land-ExM image. They are considered contact sites. Below is the revised text:

“Studying organellar contact sites is important because these regions are not just passive points of membrane proximity, but are active hubs for communication, coordination, and metabolic regulation inside cells. The region between two organelles that come into close proximity, typically separated by only 10-80 nanometers, is considered a contact site. Multi-color 3D land-ExM with lipid, protein, and antibody channels offers the specificity and resolution needed to identify organellar contact sites between organelles, including both membrane and membraneless organelles. We measure the distance between the membranes of two organelles or between a membrane and the edge of a phase-separated organelle as a metric to discriminate a contact site. As a demonstration, we investigated stress granules (SGs),...”

In the revised main text, we defined the “phase separation” in section *Land-ExM identifies membrane organelles and phase separation based on morphology*:

“Land-ExM can identify membrane structures and phase-separated structures based on their morphologies and locations in both protein and lipid channels. Here, we define phase separation as membraneless protein condensates. Fig. 3A-G display a gallery of land-ExM protein images of phase-separated organelles, such as nucleoli (Fig. 3A), ...”

3. Tunnels through the nucleus/deep invaginations of the nuclear envelope have been described in many prior studies including from light and electron microscopy and these studies should be cited - indeed there are whole reviews on this topic including PMIDs 21514163 and 39367888). I also would suggest that the authors use the term tunnel if they pass all the way from one side to the other (usually between basal to apical surfaces of the nucleus) versus an invagination, which does not. Given the whole volumes they can visualize this distinction should be possible.

We acknowledge that the nuclear tunnel-nucleolus interaction was reported as early as 70's using electron microscopy. We cited the review (<https://doi.org/10.1016/j.tcb.2011.03.008>) and original paper ([https://doi.org/10.1016/S0022-5320\(79\)90165-5](https://doi.org/10.1016/S0022-5320(79)90165-5)) in the 4th paragraph of the Result section:

“With this resolution and land-ExM's high signal-to-noise ratio, contact sites between the nuclear tunnel and the nucleolus were clearly visualized (Fig. 1K&L). This observation is consistent with previous findings using electron microscopy (Bouteille and Hemon, 1979; Malhas et al., 2011). Compared with electron microscopy, land-ExM's faster speed, 3D imaging, and multiplexity will enable a more statistical understanding of the interactions between the nuclear tunnel and the nucleolus. We explored the frequency and functions of the nuclear tunnel-nucleolus interaction with land-ExM in a recent preprint (Zhuang et al., 2024).”

And we thank the reviewer for the accurate terminology. We have replaced all “nuclear invaginations” with “nuclear tunnels” in the revised manuscript.

4. If the authors wish to infer an important functional connection between the nuclear tunnel and nucleoli

then it is imperative that they show a statistical over-representation of their proximity. Even if this is the case, the data as presented do not demonstrate a "membrane-phase separation contact site" - this is a possibility but based on the evidence presented it is an over-interpretation. Indeed, it could be that the tunnel needs to deform around the nucleolus, giving the impression of a specific interaction. More generally, the authors suggest that land-ExM is generally an approach that can reveal the relationship between phase condensates and membranes. While this is potentially the case, orthogonal approaches would certainly be required to make a strong argument for such interfaces. Another obvious question is whether these structures are observed in other cell types (also the observations presented in Fig. 3) - given the literature on such channels this is likely (e.g. A549 lung adenocarcinoma cells) - but it is unclear (at least to my knowledge) whether such structures are observed in non-transformed cells.

We thank the reviewer for insights into the biological consequences of the nuclear tunnel-nucleolus contact. This is the focal point of another project in our lab. Our preprint showed that nuclear tunnels regulate ribosome biogenesis via contacting nucleolus from different perspective, including live cell imaging, nanopillar-engineered nuclear tunnels, gene perturbation (<https://doi.org/10.1101/2024.06.21.597078>). In this work, we focused on stress granule, and demonstrate the capability of land-ExM's in identify multi-organelle contact around the stress granule. We refocused the revised manuscript on method development and have de-emphasized the biological conclusions.

In response to the question of whether these structures (presumably nuclear tunnel-nucleolus contact) are observed in other cell types, we examined several cell lines (U2OS, HEK293T, MCF-10A, MDA-MB-468, MCF-7, MDA-MB-231), primary cells (HGPS patient derived, normal aging cells), and mouse tissues (TNBC PDX). This contact is a general spatial relationship across all tested cell types and tissues. We have disclosed this information in our recent preprint (<https://doi.org/10.1101/2024.06.21.597078>).

5. What element(s) of the cytoskeleton can the authors discern using land-ExM? The general term "cytoskeleton" in Fig. 2G (Line 176). Given the authors' statement that their approach is compatible with IF, the performance with regards to observing the cytoskeleton, a limitation of other approaches, would be of importance to many potential users of the method.

We stained the sample with FITC-phalloidin and used anti-FITC antibody to retain the phalloidin signals in gel. The results confirmed that F-actin is the one of the elements in the "cytoskeleton" image shown. We included this in our new supplementary Fig. S6, and in the revised main text "*A contributor to the cytoskeleton structure in the protein channel is actin, which was confirmed with phalloidin staining (Fig. S6).*"

6. While the images in Figure 3 are very provocative and possibly were only observed because of the use of land-ExM, it is imperative to understand the frequency of the relationship between G3BP1 and these nuclear channels in order to be confident in the robustness of the observation given that the Figure only features a single example.

We agree with the reviewer that it is imperative to understand the frequency of the relationship between this G3BP1 and nuclear channels. During revision, we examined 114 tunnels from more than 20 cells stressed by sodium arsenite. We found 83% tunnels contain stress granules in the cells treated with sodium arsenite. The statistic results are summarized in panels (R and S) in the Fig. 4, shown below. We now have more confident confidence in the robustness of the observation.

Fig. 4R&S (R) Pie chart of nuclear tunnels with or without SGs. Total tunnels analyzed: 114. (S) Pie chart of SG-filled nuclear tunnels that contact nucleoli versus those that do not. Total tunnel analyzed: 83.

7. A related criticism to Point 6 is in regards to the relationship between the nuclear tunnels. Although the accumulation of G3BP1 is indeed exciting, given G3BP1 participates in other functions it would be necessary to define whether these are in fact stress granules. More over, single stress granules should be readily observed in the mid-plane of the nucleus should this be the case, additional evidence to support this point would strengthen the authors' claims. Last, it is not clear whether this observation in fact represents anything beyond the previous relationship described between the ER (which is largely functionally equivalent to the outer nuclear membrane) and SGs as described in the prior studies as cited by the authors.

In response to the reviewer's questions whether G3BP1 condensates are in fact stress granules, we performed more experiments. The results are summarized in a new supplementary figure, Fig. S7, shown below. When the cells were not treated by NaAsO₂, we did not see G3BP1 condensates in nuclear tunnels or cytosol (Fig. S7A). G3BP1 diffused across the cell. When we stressed the cells by incubating cells in NaAsO₂ solution, G3BP1 condensates showed up in both nuclear tunnels and cytosol (Fig. S7B and C). When we removed the NaAsO₂, the G3BP1 condensates disappeared from nuclear tunnels and cytosol. These results validate that the G3BP1 condensates are caused by stress, and G3BP1 can be used as a stress granule marker, which is consistent with how NaAsO₂ was used in previous research to introduce stress granules (<https://doi.org/10.7554/eLife.18413>). Furthermore, we quantitatively examined if the NaAsO₂ treatment alters the nuclear tunnels. The change in the number of tunnels and their diameters before and after NaAsO₂ treatment is insignificant (Fig. S7H and I). We have clarified the experiment in the revised text.

Figure S7. Land-ExM reveals stress granules at different locations of cells.

(A-C) Land-ExM images of U2OS cells untreated or treated with NaAsO₂ for 20 min or 60 min, then immunostained with anti-G3BP1 antibody. Scale bar: 5 μm in pre-expansion unit. Linear expansion factor: 4.

(D-F) Land-ExM images of U2OS cells stained with mCLING (magenta), NHS ester dye (cyan), and immunostained with anti-G3BP1 (yellow) and anti-Sec61b (white) antibodies. Cells were untreated or treated with NaAsO₂ for 20 min or 60 min. Scale bar: 5 μm in pre-expansion unit. Linear expansion factor: 4.

(G) Magnified images of (E) showing stress granules (SG) formed adjacent to ER (orange arrowheads). Scale bar: 1 μm in pre-expansion unit.

(H) Analysis of the number of nuclear tunnels per cell with or without 60 min NaAsO₂ treatment. Each bar represents the mean ± standard error of more than 18 cells. The ns indicates $p > 0.05$ by Welch's t-test.

(I) Analysis of the diameter of nuclear tunnels in cells with or without 60 min NaAsO₂ treatment. Each bar represents the mean ± standard error of more than 20 cells. ns indicates p>0.05 by Welch's t-test. All images were taken with an Airyscan microscope.

Last, we think the formation of stress granules in nuclear tunnels represents the same understanding of ER and SGs in the cytosol described in the prior studies and cited by us. However, having the stress granule so close to the nucleolus may cause more effective capture of ribosomes and mRNAs to repress mRNA translation and protect cells from stress. This is our hypothesis. Further experiments are needed to validate this, which is beyond the scope of this method paper.

8. This point of the authors was puzzling: "Intriguingly, we also observed the endoplasmic reticulum (ER) located between stress granules (SGs), nuclear invaginations, and nucleoli. In our four-color land-ExM images, which captured lipids, proteins, and immunostained SG and ER markers (Fig. 3L), the ER was found to overlap with the membrane of nuclear tunnels and adjacent to SGs (Fig. 3M-P)" (Lines 226-229). The outer nuclear membrane, which would necessarily line such nuclear channels, has largely the same composition as ER and will therefore contain Sec61. Therefore the "nuclear invaginations" and "ER" are expected to be one and the same unless the authors can show the presence of additional membrane bilayers, which do not appear to be visualizable in the lipid staining in Fig. 3 M-Q. Therefore I do not think that the discussion in Lines 230-238 is appropriate. An estimate of the prevalence beyond this one example is also not reported.

We agree with the reviewer that ER is expected to be a part of the outer nuclear membrane. We have deleted the quoted description and changed it to "*We also observed ER overlapped with the nuclear membrane of nuclear tunnels and adjacent to SGs.*"

9. The authors state "Fig. 4 demonstrates how land-ExM revealed the MCSs between mitochondria and nuclear invaginations, a phenomenon that has been rarely reported" (Lines 262-263). First, there are many reports of contact sites between the outer nuclear membrane/nuclear envelope and mitochondria, and these should be cited (as just one recent example published in JCB see PMID 34694322). Second, the observation is that mitochondria are close apposed to the nuclear channels (at least in this one example - again there is no estimate of the prevalence in a population of cells), but it unclear both what the meaning of this observation may be nor whether this is an observation that could not be made conventional light microscopy?

We over-interpreted Figure 4 in the original manuscript. We have deleted this figure and the whole section of *Visualizing membrane contact sites*.

Minor Points:

1. I would not consider nuclear invaginations/tunnels to be organelles - I think that the authors need to use a different term if they want to include these structures (Line 86).

We changed "organelles" to "structures".

2. Line 50-51. Clarification is needed for this sentence as the meaning of the "fold" value is not clear - this should provide a clear "effective" resolution. "ExM enables light microscopes to achieve 3- to 20-fold effective resolution" - presumably this is compared to the diffraction limit in conventional light microscopy?

The fold change is compared with the before-expansion. We rephrased "By physically expanding cells or tissues, ExM enables light microscopes to achieve effective resolution" as follows:

"By physically expanding cells or tissues, ExM allows light microscopes to achieve an effective resolution that is 3- to 20-fold higher than before expansion"

3. While G3BP1 is indeed well known for its association with stress granules, it also has roles at membranous organelles, for example damaged lysosomes. Given the arsenite treatment to induce stress granules, it is possible that this localization is related to these prior studies, which should be cited and discussed.

We appreciate this insight in G3BP1's role in membrane repair! We discussed this topic at the end of the revised Discussion section as follows:

While G3BP1's classic role involves SG assembly, it also contributes to membrane repair (King et al., 2025). For example, G3BP1 is reported to repair lysosomes through a "plugging" mechanism (Bussi et al., 2023). This repair activity positions G3BP1 as a crucial integrator of cellular stress responses across both membraneless and membrane-bound compartments. It is possible that the localization of G3BP1 in the nuclear tunnels is related to these prior studies.

November 30, 2025

RE: JCB Manuscript #202502035R-A

Xiaoyu Shi
University of California, Irvine

Dear Dr. Shi,

Thank you for submitting your revised manuscript entitled "Landscape Expansion Microscopy Reveals Interactions between Membrane and Phase-Separated Organelles." The manuscript was re-reviewed by the three original reviewers. As you can see from the attached reviews, all reviewers are in principle satisfied with the revisions, with the exception of two minor issues. Before we can formally accept the manuscript, we ask that you address these remaining points raised by Reviewer 3 by revising the manuscript text, including in the abstract. Please also make any changes that are necessary to meet our formatting guidelines (see details below).

A. MANUSCRIPT ORGANIZATION AND FORMATTING:

- 1) Text limits: Character count for Tools is < 40,000, not including spaces. Count includes title page, abstract, introduction, results, discussion, and acknowledgments. Count does not include materials and methods, figure legends, references, tables, or supplemental legends.
- 2) Figure formatting: Tools may have up to 10 main text figures. Scale bars must be present on all microscopy images, including inset magnifications. Also, please avoid pairing red and green for images and graphs to ensure legibility for color-blind readers. If red and green are paired for images, please ensure that the particular red and green hues used in micrographs are distinctive with any of the colorblind types. If not, please modify colors accordingly or provide separate images of the individual channels.
- 3) Statistical analysis: Error bars on graphic representations of numerical data must be clearly described in the figure legend. The number of independent data points (n) represented in a graph must be indicated in the legend. Please indicate whether 'n' refers to technical or biological replicates (i.e. number of analyzed cells, samples or animals, number of independent experiments). If independent experiments with multiple biological replicates have been performed, we recommend using distribution-reproducibility SuperPlots (please see Lord et al., JCB 2020) to better display the distribution of the entire dataset, and report statistics (such as means, error bars, and P values) that address the reproducibility of the findings.

Statistical methods should be explained in full in the materials and methods. For figures presenting pooled data the statistical measure should be defined in the figure legends. Please also be sure to indicate the statistical tests used in each of your experiments (both in the figure legend itself and in a separate methods section) as well as the parameters of the test (for example, if you ran a t-test, please indicate if it was one- or two-sided, etc.). Also, if you used parametric tests, please indicate if the data distribution was tested for normality (and if so, how). If not, you must state something to the effect that "Data distribution was assumed to be normal but this was not formally tested."
- 4) Materials and methods: Should be comprehensive and not simply reference a previous publication for details on how an experiment was performed. Please provide full descriptions (at least in brief) in the text for readers who may not have access to referenced manuscripts. The text should not refer to methods "...as previously described."
- 5) For all cell lines, vectors, strains, constructs/cDNAs, etc. - all genetic material: please include database / vendor ID (e.g. Addgene, ATCC, etc.) or if unavailable, please briefly describe their basic genetic features, even if described in other published work or gifted to you by other investigators (and provide references where appropriate). Please be sure to provide the sequences for all of your oligos: primers, si/shRNA, RNAi, gRNAs, etc. in the materials and methods. You must also indicate in the methods the source, species, and catalog numbers/vendor identifiers (where appropriate) for all of your antibodies, including secondary. If antibodies are not commercial, please add a reference citation if possible.
- 6) Microscope image acquisition: The following information must be provided about the acquisition and processing of images:
 - a. Make and model of microscope
 - b. Type, magnification, and numerical aperture of the objective lenses
 - c. Temperature
 - d. Imaging medium

- e. Fluorochromes
- f. Camera make and model
- g. Acquisition software
- h. Any software used for image processing subsequent to data acquisition. Please include details and types of operations involved (e.g., type of deconvolution, 3D reconstitutions, surface or volume rendering, gamma adjustments, etc.).

7) References: There is no limit to the number of references cited in a manuscript. References should be cited parenthetically in the text by author and year of publication. Abbreviate the names of journals according to PubMed.

8) Supplemental materials: Tools generally may have up to 5 supplemental figures and 10 videos. You currently exceed this limit but, in this case, we will be able to give you the extra space. Please also note that tables, like figures, should be provided as individual, editable files. A summary of all supplemental material should appear at the end of the Materials and methods section. Please include one brief sentence per item.

9) eTOC summary: A ~40-50 word summary that describes the context and significance of the findings for a general readership should be included on the title page. The statement should be written in the present tense and refer to the work in the third person. It should begin with "First author name(s) et al..." to match our preferred style.

10) Conflict of interest statement: JCB requires inclusion of a statement in the acknowledgements regarding competing financial interests. If no competing financial interests exist, please include the following statement: "The authors declare no competing financial interests." If competing interests are declared, please follow your statement of these competing interests with the following statement: "The authors declare no further competing financial interests."

11) A separate author contribution section is required following the Acknowledgments in all research manuscripts. All authors should be mentioned and designated by their first and middle initials and full surnames. We encourage use of the CRediT nomenclature (<https://casrai.org/credit/>).

12) ORCID IDs: ORCID IDs are unique identifiers allowing researchers to create a record of their various scholarly contributions in a single place. Please note that ORCID IDs are required for all authors. At resubmission of your final files, please be sure to provide your ORCID ID and those of all co-authors.

13) Journal of Cell Biology now requires a data availability statement for all research article submissions. These statements will be published in the article directly above the Acknowledgments. The statement should address all data underlying the research presented in the manuscript. Please visit the JCB instructions for authors for guidelines and examples of statements at (<https://rupress.org/jcb/pages/editorial-policies#data-availability-statement>).

B. FINAL FILES:

Thank you for your attention to these final processing requirements. Please revise and format the manuscript and upload materials within 7 days. If you need an extension for whatever reason, please let us know and we can work with you to determine a suitable revision period.

Thank you for this interesting contribution, we look forward to publishing your paper in Journal of Cell Biology.

Sincerely,

Joerg Bewersdorf, PhD
Monitoring Editor
Journal of Cell Biology

Dan Simon, PhD
Scientific Editor
Journal of Cell Biology

Reviewer #1 (Comments to the Authors (Required)):

The authors have addressed Reviewers' concerns.

Reviewer #2 (Comments to the Authors (Required)):

The authors have addressed my comments sufficiently.

Reviewer #3 (Comments to the Authors (Required)):

The revised manuscript is improved in several aspects including comparisons of Land-ExM to other expansion/labeling approaches and more quantitative metrics.

However, I still feel that two aspects of the study are over-stated or are misinterpreted.

First, I remain unconvinced that any static imaging approach can "identify...phase-separated structures based on their morphologies and locations in both protein and lipid channels." (Line 275). If the authors are suggesting that they can see the details of a structure that is thought to be phase-separated based on rigorous prior examination including live-cell approaches in other work, that would be reasonable. However, the title of Figure 3 (Land-ExM identifies phase-separated and membrane organelles) cannot be justified - because it suggests that Land-ExM could make such a determination on its own.

Second, there remains one crucial misunderstanding related to Figure 4. The authors continue to state that "We also observed ER overlapped with the nuclear membrane of nuclear tunnels and adjacent to SGs" with the Figure 4 title being: "The nuclear tunnel forms a quadruple-organellar contact site that includes the stress granule, the ER, the nucleolus, and itself." (Line 381). The nuclear membrane is composed of an inner and outer nuclear membrane. Thus, the tunnel has an inner and outer nuclear membrane. The outer nuclear membrane is, in composition, ER. Thus, ER does not "overlap" with the nuclear membrane - rather the outer nuclear membrane and ER are one and the same. Thus, as raised previously, while there may be a triple-organellar contact site between the nucleolus, the nuclear envelope (inner and outer nuclear membranes) and SGs, as the nuclear tunnel and ER (Sec61 staining) are the same structure, it cannot be called a quadruple-organellar contact site.